# Adeno-Associated Virus 2 (AAV2) - Induced RPA exhaustion generates cellular DNA damage and restricts viral gene expression

Monnette F. Summers[1,2], MegAnn K. Haubold[2], Marcel Morgenstern[3,4], Phoenix Shepherd[1,5,6], Clairine I. S. Larsen[1,2], Ava E. Bartz[1,2], Gopishankar Thirumoorthy[7], Robert N. Kirchdoerfer[1,5,6], Joshua J. Coon[3,4,8,9], Kavi P. M. Mehta[7,10], Kinjal Majumder[1,2,10]*

1 Institute for Molecular Virology, University of Wisconsin-Madison, Madison, Wisconsin, United States of America, 2 McArdle Laboratory for Cancer Research, School of Medicine and Public health, Madison, Wisconsin, United States of America, 3 Department of Biomolecular Chemistry, School of Medicine and Public health, Madison, Wisconsin, United States of America, 4 National Center for Quantitative Biology of Complex Systems, University of Wisconsin-Madison, Madison, Wisconsin, United States of America, 5 Department of Biochemistry, University of Wisconsin-Madison, Madison, Wisconsin, United States of America, 6 Center for Quantitative Cell Imaging, University of Wisconsin-Madison, Madison, Wisconsin, United States of America, 7 Department of Comparative Biosciences, School of Veterinary Medicine, Madison, Wisconsin United States of America, 8 Department of Chemistry, University of Wisconsin-Madison, Madison, Wisconsin, United States of America, 9 Morgridge Institute for Research, Madison, Wisconsin, United States of America, 10 University of Wisconsin Carbone Cancer Center, Madison, Wisconsin, United States of America

* kmajumder@wisc.edu

## Abstract

Parvoviruses are single-stranded DNA viruses that have been modified to serve as vehicles for therapeutic transgene delivery in the form of recombinant Adeno-Associated Virus (rAAV2) vectors or rodent parvovirus-derived oncolytic agents. Infection with viruses of the *Parvoviridae* family induces a cellular DNA Damage Response (DDR) signal that supports virus replication. However, it remains unknown whether rAAV2 vectors or non-replicative wild-type AAV2 (wtAAV2) genomes induce cellular DDR signals, which might be deleterious to the cell. To determine the impact of wtAAV2/rAAV2 genomes on the integrity of the host chromosome, we have pulsed wtAAV2/rAAV2 infected cells with BrdU analogs followed by single-molecule imaging of the cellular replisomes and proteomic analysis of the host replication forks. We discovered that non-replicative wtAAV2/rAAV2 genomes are sufficient to induce replication stress on the host genome, leading to DDR signals in a dose-dependent manner. Moreover, infection with replication-competent wtAAV2 leads to enrichment of replication stress proteins, DNA repair factors and RNA processing machinery on cellular replication forks. However, neither the wtAAV2 Inverted Terminal Repeats (ITRs) that are retained in rAAV2s nor empty capsids are sufficient to induce host-cell replication stress. Strikingly, incoming wtAAV2 genomes associate with the single-stranded DNA binding protein RPA in host cells in a dose-dependent manner, progressively

**Data availability statement:** The mass spectrometry data of iPOND-MS analysis has been uploaded to MassIVE proteomics repository and can be accessed with the accession number MSV000098258. https://massive.ucsd.edu/ProteoSAFe/static/massive.jsp

**Funding:** This research was funded partially by NIH/NIAID K99/R00 Pathway to Independence Award, grant number AI148511 to K.M.; Wisconsin Partnership Program's New Investigator Award (PERC Grant G-4942) to K.M.; NIH/NIGMS R35 Maximizing Investigator's Research Award (MIRA), grant number GM154938 to K.M and NIH P41GM108538 to J.C. C.I.S.L. is funded by an NSF Graduate Research Fellowship Program award 2137424 and M.K.H by an NSF Graduate Research Fellowship Program award 2023348714. M.F.S. is funded by a SciMED Graduate Research Scholarship from the University of Wisconsin-Madison and Molecular and Cellular Pharmacology T32 training grant T32GM141013 from the NIH. The funders had no role in study design, data collection and analysis, decision to publish, or preparation of the manuscript.

**Competing interests:** The authors have declared that no competing interests exist.

shortening cellular replication forks. These elevated levels of wtAAV2-induced cellular replication stress eventually leads to accumulation of DDR signals in the nucleus. Chemical inhibition of RPA activity and RNAi-mediated knockdown leads to de-repression of the wtAAV2 genome, increasing Rep 68/78 gene expression. Ectopic expression of RPA rescues wtAAV2-induced replication stress. Taken together, our findings suggest that depletion of cellular stores of RPA molecules by competing wtAAV2 genomes restrict viral gene expression and cause cellular DNA damage.

## Author summary

Adeno-Associated Viruses 2 (wtAAV2) are modified to design therapeutic gene therapy vectors, but how they interact with the guardians of host DNA remains unknown. In this work, we show that wtAAV2 genomes compete with the host cell for the single-stranded DNA binding protein RPA, rendering the host vulnerable to replication stress leading to both suppression of the viral gene expression and induction of cellular DNA breaks. These findings provide insights into how gene therapies delivered at high doses could have genotoxic effects, underscoring the importance of engineering wtAAV2-based gene therapy platforms that express efficiently at lower doses.

## Introduction

DNA breaks generated by viral and non-viral agents activate DNA Damage Response (DDR) signals that have evolved to ensure the fidelity of the genetic code. These signals facilitate DNA repair by recruiting cellular DNA processing factors that shut down transcription in the vicinity of the DNA break, process the DNA strand and restore the integrity of the genome. Primarily regulated by the evolutionarily conserved Phosphatidylinositol-3-Kinase-like kinases ATM, ATR and DNAPK, these DDR signals are usurped by viruses to regulate the outcome of viral infection. To carry out their life cycles in the nuclear compartment, DNA viruses either trigger (parvoviruses [1,2], polyomaviruses [3,4] and papillomaviruses [5,6]) or inactivate (adenovirus [7–11] and herpesviruses [12]) cellular DDR signaling pathways for distinct infectious outcomes (such as the establishment of latency [12] or production of progeny virus [13–15]). DNA viruses interfere with DDR signaling pathways activated by ATM, ATR or DNA-PK signals using their viral proteins (adenovirus) or nucleic acids (herpesviruses and adenoviruses). Parvoviruses are single-stranded DNA viruses that can replicate in cells in S-phase (referred to as autonomous parvovirus) or with the help of co-infecting viruses (known as dependoparvovirus). Replication of the dependoparvovirus Adeno-Associated Virus Type 2 (wtAAV2) induces a cellular DDR signal using the DNA-PK pathway [16], but the signals associated with non-replicative (lacking a co-infecting virus; hereafter referred to as mono-infection) wtAAV2 infection remain largely unknown.

The cellular ATR signaling pathway has evolved to respond to single-stranded DNA breaks in the nucleus that are generated by replication stress and transcription-replication conflicts [17,18]. Viruses deploy a diverse set of programs to dysregulate ATR-mediated signals for their benefit (reviewed in [17]). Rift Valley Fever Virus and the autonomous parvovirus Minute Virus of Mice (MVM), for example, express proteins that inactivate components of the ATR pathway [19,20]. Papillomavirus E2 proteins and Polyomavirus Large-T antigen interact with ATR-pathway proteins (TOPBP1 [21] and Claspin [22] respectively) and components of eukaryotic replisomes to control ATR-dependent signals. The genomes of the autonomous parvovirus Minute Virus of Mice (MVM) sequester the cellular stores of Replication Protein A (RPA), the principal sensor of single stranded DNA in the nucleus, which in turn initiates ATR signaling, thereby rendering the host genome vulnerable to replication stress. This culminates in the generation of extensive cellular DNA damage [23], contributing to the induction of a potent pre-mitotic cell cycle arrest at the G2/M border [24], an outcome that has been leveraged to engineer oncolytic virotherapies using protoparvoviruses [25]. Since MVM replication rapidly generates single-stranded DNA, double stranded DNA and RNA-DNA hybrid intermediates [26], it remains unknown what aspect of viral replication activates cellular DDR signals. In the absence of co-infecting "helper" viruses, non-replicative wild-type Adeno-Associated Viruses type 2 (wtAAV2) and the recombinant AAV2 (rAAV2) gene therapy vectors that are derived from them represent a tractable system to dissect the DDR signals activated by viral infection [10].

We have previously discovered that genomes of autonomous parvoviruses like MVM and dependoparvoviruses like wtAAV2 localize to cellular sites of DNA damage [2,27]. At these sites, MVM forms replication centers called Autonomous Parvovirus Associated Replication (APAR) bodies [28,29], whereas wtAAV2/rAAV2 genomes form subnuclear structures where their genomes are converted from single-stranded to double-stranded forms [30]. MVM-generated subnuclear structures are formed in proximity to large host chromatin domains spanning multiple megabases [2,31] but AAV2-associated nuclear structures associate with distinct regions that colocalize with sharp chromatin sites that are in the vicinity of DNA breaks [27]. Many of these cellular DDR sites are fragile genomic regions [32–34] where the cellular replication and transcription machineries collide to form transient secondary structures that recruit DNA processing factors (such as ATR), serving as fertile milieu for establishment and replication of viruses in their vicinity [35–38]. ATR-mediated signals regulate the localization to cellular DDR sites of the parvovirus non-structural protein NS1 (for MVM) and Rep 67/78 (for wtAAV2) and are essential for efficient virus replication [27,39]. At late stages of viral life cycle, MVM inactivates the ATR signaling pathway by sequestering Casein Kinase 2 (CK2), blocking its ability to phosphorylate downstream substrates [40]. However, it remains unknown whether ATR signaling is altered during wtAAV2 mono-infection.

The outcome of wtAAV2 mono-infection on host DDR pathways remains controversial. Some of these prior studies posit that wtAAV2 genomes persist in the nuclear compartment as a molecular mimic of stalled host replication forks [41], leading to local activation of CHK1-related signals (downstream of ATR activation). These CHK1-dependent signals cause cell death in cells that lack p53 signaling [42,43]. On the other hand, cellular DDR signals associated with double stranded breaks monitored by phosphorylation of H2AX (referred to as γH2AX) are not always detected in wtAAV2/rAAV2-infected host cells [44,45]. In contrast, transduction of human Embryonic Stem (ES) cells with rAAV2 leads to host genome instability and p53-dependent cell death [46]. Nevertheless, it is incontrovertible that parvoviruses, lacking polymerases of their own, rely exclusively on the host for polymerases to amplify their genomes. Cellular DDR signaling pathways regulate the licensing of replicative DNA polymerases alpha, delta and epsilon [47–49]; as well as translesion polymerases eta, zeta and kappa [50]. While inhibition of DNA polymerases alpha and delta inhibits the replication of Parvovirus B19V [51], eta and kappa knockdowns decrease Human Bocaparvovirus (HBoV1) replication [52]. Though the absence of polymerase eta and kappa reduces wtAAV2 replication [44], this occurs only in the presence of adenovirus "helper" proteins (E1, E2, E4orf6 and VA-RNA [53,54]). Importantly, inhibition of ATR signaling arrests wtAAV2 replication [44], possibly because ATR regulates polymerase eta and kappa function [55,56]. Since repair polymerases must be licensed by host activation signals [57], these observations additionally suggest that wtAAV2 must activate local signals that are yet to be elucidated. However, wtAAV2 mono-infection (i.e., infection in the absence of a co-infecting helpervirus) does not support viral

genome amplification, eliminating the need of cellular DDR signals to license virus replication, suggesting that cellular DDR signals are generated as a host response to viral infection and not as a means of viral pathogenesis.

In this study, we have investigated whether and how wtAAV2 mono-infection impacts the stability of the eukaryotic replisome using assays that label the nascent host DNA strand. Imaging of the eukaryotic replisomes and mass-spectrometry-based identification of the proteins associated with replicating DNA revealed an enrichment of replication-stress markers upon wtAAV2 infection. This is caused by the AAV2 genomes serving as molecular decoys for the single-stranded DNA binding protein RPA, rendering the host genome vulnerable to replication stress. Surprisingly, while wtAAV2-induced RPA exhaustion leads to cellular DNA damage at high doses, the binding of RPA molecules to wtAAV2 restricts viral gene expression. Taken together, our findings define how wtAAV2 genomes modulate the replication fork stability in host cells that can cause genomic DNA damage, explaining how viral genes are repressed by replication stress proteins in the absence of coinfecting helperviruses.

## Results

### Replicative and non-replicative wtAAV2 genomes induce replication stress in cycling cells

Parvovirus replication induces cellular DDR signals monitored by increase in γH2AX levels and comet tails associated with host DNA fragmentation [1,2,44], but it remains unclear whether mono-infection induces cellular DNA damage. Consistent with this model, wtAAV2 replication in the presence of helper viruses (Adenovirus or Herpesvirus) or ectopically expressed Adenovirus helper proteins induces a robust cellular DDR but the impact of wtAAV2 mono-infection on the cellular DDR pathways is much less pronounced. Since the wtAAV2 genome mimics a stalled-fork [41], we hypothesized that these entities compete with the host genome for cellular replication-stress factors. To determine whether wtAAV2 mono-infection impacts the stability of host cell replication forks, we utilized single-molecule DNA Fiber Assays (DFAs; [58]). DFAs measure the length of host cell replication tracts using sequential pulses of the halogenated nucleoside analogs Iododeoxyuridine (IdU) and Chlorodeoxyuridine (CldU) followed by spreading of DNA fibers on positively charged slides analyzed by confocal imaging. As schematized in Fig 1A, DFAs of wtAAV2-infected HEK293T cells at 24 hours post-infection (hpi) revealed a significant shortening of cellular replication forks (representative examples in Fig 1B) during wtAAV2 mono-infection from CldU lengths of 4.1 μm in Mock-infected cells to 3.1 μm in wtAAV2-infected cells. Interestingly, cells transfected with only the pHelper plasmid (expressing the adenovirus helper proteins E2A, E4orf6 and VA-RNA) are sufficient to shorten host-cell replication forks (measured by both IdU and CldU labels, Fig 1C,D). To validate these observations in independent cell line models of wtAAV2 infection, we performed DFAs in A549 (airway epithelial) and U2OS (osteosarcoma) cells. We discovered that upon infection with wtAAV2 for 24 hours at an MOI of 5,000 vg/cell (schematized in S1A Fig), both cell types exhibited shortened cellular replication forks (S1B and S1C Fig). It is important to note, however, that DFAs represent an average of multiple cells under infection conditions. To validate our observation of replication stress induction at a single-cell level using an independent technique, we performed wtAAV2 infection of U2OS osteosarcoma cells and monitored the ability of host replisomes to incorporate 5-Ethynyl-2'-deoxyuridine (EdU) analogs by conjugating fluorophores using Click-iT chemistry. Infection of U2OS cells with wtAAV2 for 24 hours at a multiplicity of 5,000 viral genomes per cell led to fewer and smaller EdU-positive foci (representative image in Fig 1E and quantified in Fig 1F). The median number of EdU-positive foci in wtAAV2-infected cells decreased to 3 from 6 in mock infected cells (Fig 1F), phenocopying our DFA analysis in 293T, U2OS and A549 cells. To determine when wtAAV2 mono-infection induces cellular replication stress, we performed DFAs at 12 hpi, 24 hpi and 36 hpi. As shown in Fig 1G (IdU) and 1H (CldU), there was no robust change in median fork lengths from the onset of infection to 12 hpi, indicating that wtAAV2-induced replication stress is not an acute response. However, there was a substantial decrease in median fiber lengths between 12 hpi and 24 hpi, which plateaued by 36 hpi (Fig 1G,H). This plateau of wtAAV2-induced replication fork shortening at 36 hpi suggested that the host cell might deploy mechanisms to overcome the virus-induced replication stress. Taken together, our observations suggested that wtAAV2 mono-infection induces replication stress on the host genome.

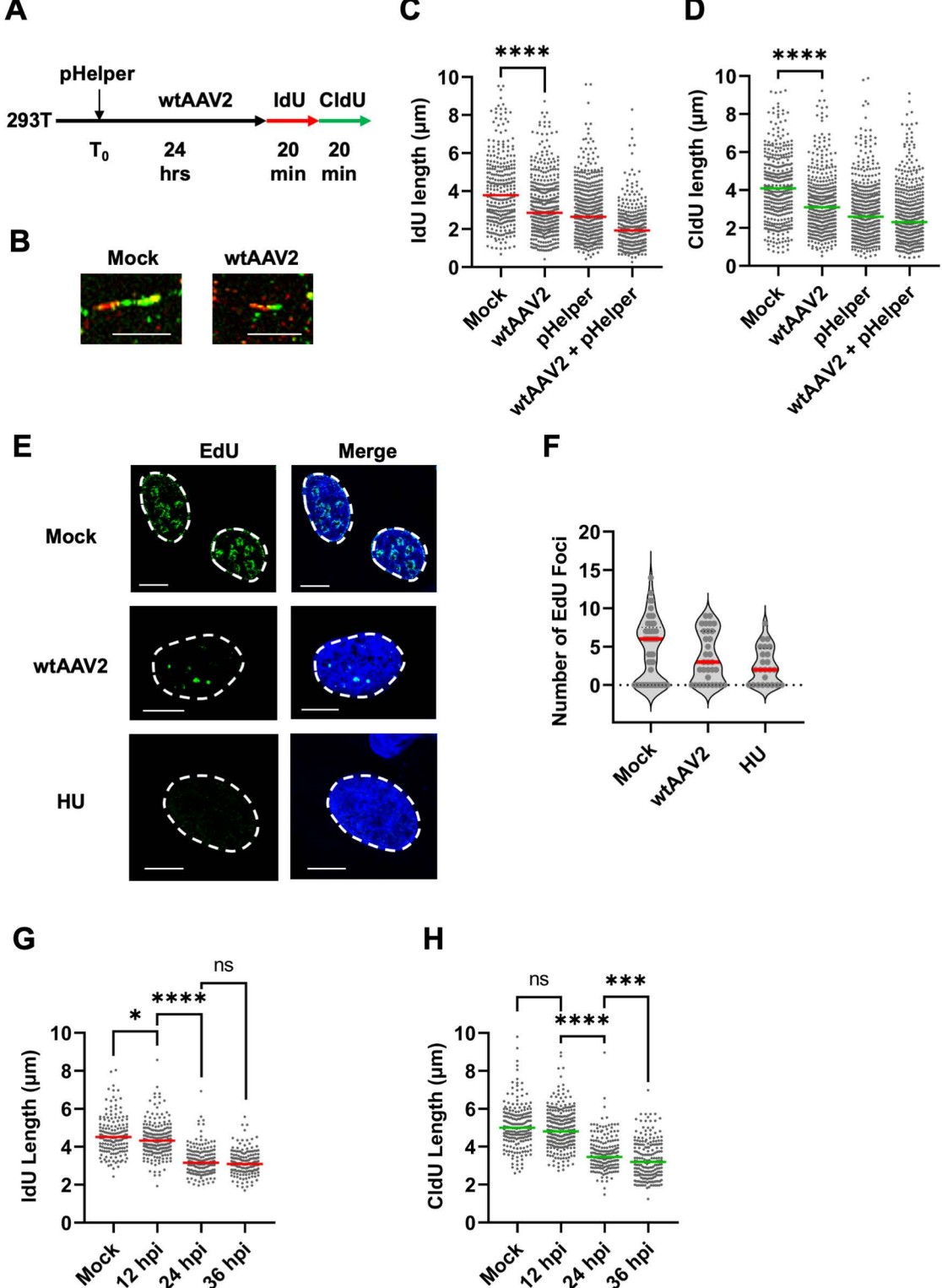

**Fig 1. Replicative and non-replicative wtAAV2 genomes induce replication stress in cycling cells.** (A) Schematic of the timeline of wtAAV2 infection of 293T cells followed by sequential pulsing with IdU and CldU for DNA fiber analysis. (B) Representative images of single DNA fibers in mock (left) compared with wtAAV2-infected 293T cells (right) at 24 hpi. The red portions of the fibers represent IdU label and green portions represent CldU

PLOS Pathogens

labels. The horizontal scale bar represents 5 micrometers. (C, D) DNA fiber length measurements were indicated by each datapoint for (C) IdU-labelled fibers and (D) CldU-labelled fibers. The median lengths of many measurements of IdU and CldU labelled fibers are represented by red and green bars respectively. At least 150 individual fibers were measured for each condition for three independent replicates of AAV2 infections of 293T cells. Statistical significance was measured by Mann Whitney Wilcoxon test, **** represents P < 0.0001. (E) Representative images of cellular replication forks (green) in uninfected U2OS cells compared with U2OS cells infected with wtAAV2 and treated with 2 µM hydroxyurea (HU) for 12 hours. Cells were pulsed with EdU followed by conjugation of Alexa Fluor 488 using click chemistry. The cell nucleus is stained with DAPI (blue) and the nuclear borders are demarcated with a white dashed line. The scale bar represents 5 micrometers. (F) Quantification of EdU labelling in 1E with each measurement represented by a datapoint on the graph. The median values are represented by red horizontal bars from EdU foci counted in over 30 U2OS cell nuclei. (G, H) Measurement of single molecule DNA fiber lengths in wtAAV2 infected 293T cells at the indicated timepoints of infection which were followed by sequential pulses of IdU and CldU for 20 minutes each. The data is presented as IdU lengths (G) and CldU lengths (H) with the red and green horizontal bars showing the respective median values. Statistical significance was measured by Mann Whitney Wilcoxon test, **** represents P < 0.0001, *** P < 0.001, * P < 0.05, ns represents differences that are not statistically significant.

## Recombinant AAV2 vector genomes induce replication stress in host cells

Recombinant AAV2 gene therapy vectors have been engineered from wtAAV2 by replacing the Rep-Cap containing open reading frames with therapeutic transgenes that are the under control of transcriptional regulatory elements [59]. Transduction of cells with rAAV2 vectors have previously yielded conflicting results regarding DNA damage induction, with early reports indicating that rAAV2 vectors activate CHK1 kinase globally [42], induce cell death in transformed cycling cells [43] and toxicity in human Embryonic Stem (ES) cells [46]. Surprisingly, follow-up studies found that non-replicative wtAAV2 and rAAV2 are insufficient to provoke global DDR signals [16,44,60]. To determine whether rAAV2 genomes disturb the cellular replisome, we performed DFAs in 293T cells transduced with rAAV2 at an MOI of 5,000 viral genomes per cell for 24 hours (Fig 2A). We compared the cellular replisome in rAAV2-transduced cells with that of self-complementary AAV2 (scAAV2) that contain only one wtAAV2-ITR and form a double-stranded DNA molecule soon after nuclear entry (removing the need for scAAV2s to undergo DNA processing). As shown in Fig 2B and C, both rAAV2 and scAAV2 transduction led to a shortening of cellular replication forks measured by shortening in the median lengths of both nucleoside analog (IdU and CldU). The median lengths of cellular replication forks in wtAAV2-infected cells were comparable to that of rAAV2/scAAV2-transduced (lacking the Rep and Cap ORFs) 293T cells at 24 hpi. This observation indicated that: 1) the non-coding components of wtAAV2 genomes might be involved in replication stress, 2) the conversion of single-stranded rAAV genomes into double stranded molecules is not sufficient, and 3) the low levels of Rep 68/78 protein generated by wtAAV2 mono-infection is not sufficient to account for the observed cellular replication stress. The similar levels of replication stress induced by wtAAV2, rAAV2 and scAAV2 genomes suggested that either the ITR (which is shared between at three genomic forms) or the viral particle (VP; also shared between all three infection/transductions) might be sufficient to contribute to wtAAV2-induced replication fork shortening.

## Empty wtAAV2 capsids and ITR elements are not sufficient to induce host-cell replication stress

To determine whether the wtAAV2 ITR element is sufficient to induce host-cell replication stress, we transfected 293T cells with 10,000 and 1,000 copies of a non-self-complementary ITR sequence containing oligonucleotides for 24 hours before pulsing cells with IdU/CldU for DFAs (Fig 2D). We compared the lengths of the replication forks in these cells with that of cells transfected with equivalent number of copies of pUC18 plasmid DNA and cells treated with HU. Measurement of the replication forks in these cells revealed that the ITR elements were not sufficient to cause cellular replication stress measured by IdU (Fig 2E) or CldU lengths (Fig 2F). To determine whether the wtAAV2 Capsid is sufficient to induce cellular replication stress, we transduced 293T cells with 10,000 or 1,000 copies of the empty wtAAV2 capsids for 24 hours before performing DFAs (according to the schematic in Fig 2G). As shown in Fig 2H (IdU) and 2I (CldU), transduction of these empty capsids was not sufficient to shorten the host-cell replication fibers. To determine whether heterologous DNA molecules at varying multiplicities are sufficient to induce cellular replication stress, we performed DFAs in 293T cells transfected with pUC18 for 24 hours before being pulsed for CldU incorporation (S2A Fig). As demonstrated in DFAs in

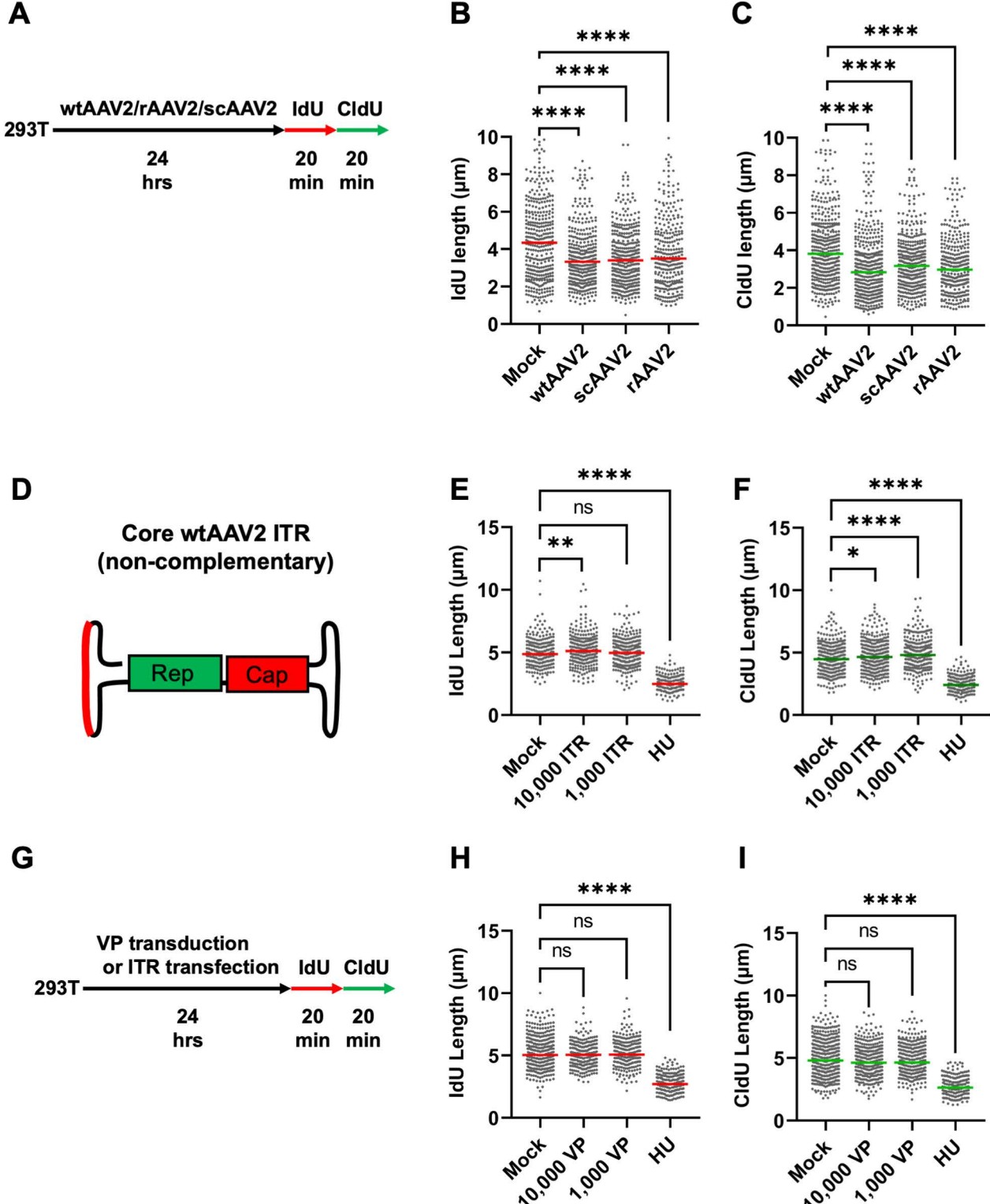

**Fig 2. Recombinant AAV2 vector genomes induce replication stress in host cells.** Schematic of the timeline of wtAAV2 infection and rAAV2/scAAV2 transduction of 293T cells followed by sequential pulsing with IdU and CldU for 20 minutes each prior to DNA fiber analysis. (B, C) DNA fiber length measurements were indicated by each datapoint for (B) IdU-labelled fibers and (C) CldU-labelled fibers. The median lengths of many

measurements of IdU and CldU labelled fibers in wtAAV2 infected cells and rAAV2/scAAV2 transduced cells are represented by red and green bars respectively. At least 150 individual fibers were measured for each condition for three independent replicates of AAV2 infections of 293T cells. Statistical significance was measured by Mann Whitney Wilcoxon test, **** P < 0.0001, *** P < 0.001, ** < 0.01, ns represents differences that are not statistically significant. (D) Schematic of the wtAAV2 genome with red line indicating the non self-complementary region of the ITR transfected into 293T cells for 24 hours prior to sequential IdU/CldU labelling for 20 mins each followed by DNA fiber analysis. (E, F) DNA fiber length measurements were indicated by each datapoint for (E) IdU-labelled fibers and (F) CldU-labelled fibers. The median lengths of many measurements of IdU and CldU labelled fibers in empty VLP transduced cells are represented by red and green bars respectively. (G) Schematic of the timeline of empty assembled VLP transduction or ITR element transfection of 293T cells followed by sequential pulsing with IdU and CldU for 20 minutes each prior to DNA fiber analysis. (H,I) Representation of the lengths of the (H) IdU and (I) CldU forks with median lengths represented by red and green bars respectively. At least 150 individual fibers were measured for each condition for three independent replicates in of 293T cells. Statistical significance was measured by Mann Whitney Wilcoxon test, **** represents P < 0.0001, ns represents differences that are not statistically significant.

S2B Fig, neither 5,000 pUC18 plasmid genomes, nor 25,000 pUC18 plasmid genomes were sufficient to induce replication stress (S2B Fig). These were also not sufficient to induce cellular DDR signals monitored by gamma H2AX levels (S2C Fig). However, transfection of the plasmid genome of AAV2 (pAAV2) into 293T cells for 24 hours was sufficient to cause a dose-dependent shortening of cellular replication forks (S2B Fig) and an associated induction of gamma H2AX signals (S2D Fig). We surmise that this induction of cellular DDR signals and replication stress might be caused by AAV2 REP 68/78 proteins that are expressed by the transfected plasmid genomes (S2D Fig). These findings suggested that the non-self-complementary ITR region, empty viral particles and heterologous double-stranded DNA molecules are not sufficient to induce cellular replication stress. A combination of these components working together with host factors likely generate host-cell replication stress.

## Characterization of the wtAAV2-induced alterations to the human replisome using iPOND-MS

To determine which host proteins are altered at eukaryotic replisomes by wtAAV2 infection and corroborate our observations of wtAAV2-induced replication stress (using DFAs and EdU labelling), we utilized the proteomics approach iPOND [Isolation of Proteins On Nascent DNA; [61]]. iPOND uses EdU-based labelling followed by Click-chemistry-mediated biotinylation coupled with streptavidin pulldowns to identify the proteins associated with nascent DNA [61]. We performed iPOND analysis in 293T and U2OS cells infected with wtAAV2 at an MOI of 10,000 vg/cell at 24 hpi. As shown in Fig 3A, iPOND-MS analysis of wtAAV2-infected cells yielded 1084 proteins that were dysregulated in U2OS cells and 1606 proteins in 293T cells. Out of these, 822 proteins were shared between U2OS and 293T cells (Fig 3A). Amongst the shared proteins that are dysregulated by wtAAV2, 414 were enriched and 408 were depleted at host-cell replication forks (Fig 3B). Unbiased pathway analysis of these 822 proteins revealed that these factors are involved in DNA replication, double-stranded DNA break repair and chromosome organization (Fig 3C). Strikingly, several proteins are also involved in RNA processing, RNA localization and splicing (Fig 3C). Host proteins enriched at cellular replication forks during infection included DNA processing factors such as GINS, TOP2A, PRIM1/2, RPA2 and XRCC1; DDR signaling proteins such as CSNK2A1/3; cell cycle regulator proteins such as TFDP2 and SFN1; as expected for cells undergoing replication stress. Host DDR-associated proteins depleted at cellular replication forks in wtAAV2-infected cells included SIK3, DAXX and NBN (Fig 3D). To determine whether these host replication fork-associated proteins regulate wtAAV2 gene expression, we performed RNAi-knockdowns in 293T cells, infected with wtAAV2 for 24 hpi and assessed their impact on Rep 68/78 transcript levels. As shown in Fig 3E, knockdowns of XRCC1, SIK3, PRIM1 and TFDP2 did not impact wtAAV2 gene expression substantially (verifications of the knockdowns are presented in S3A–S3D Fig). However, knock-down of RPA2 (also known as RPA32) led to a substantial de-repression of wtAAV2 gene expression, causing a 10-fold increase of Rep 68/78 transcripts (Fig 3E, left y-axis). Strikingly, absence of CSNK2A1/3 led to a 300-fold increase in wtAAV2 gene expression relative to Mock knockdown cells (Fig 3E, right y-axis). These observations corroborate and build on the recent findings of CSNK1 serving as a host restriction factor for scAAV2 transduction [62]. In sum, these findings orthogonally validated the

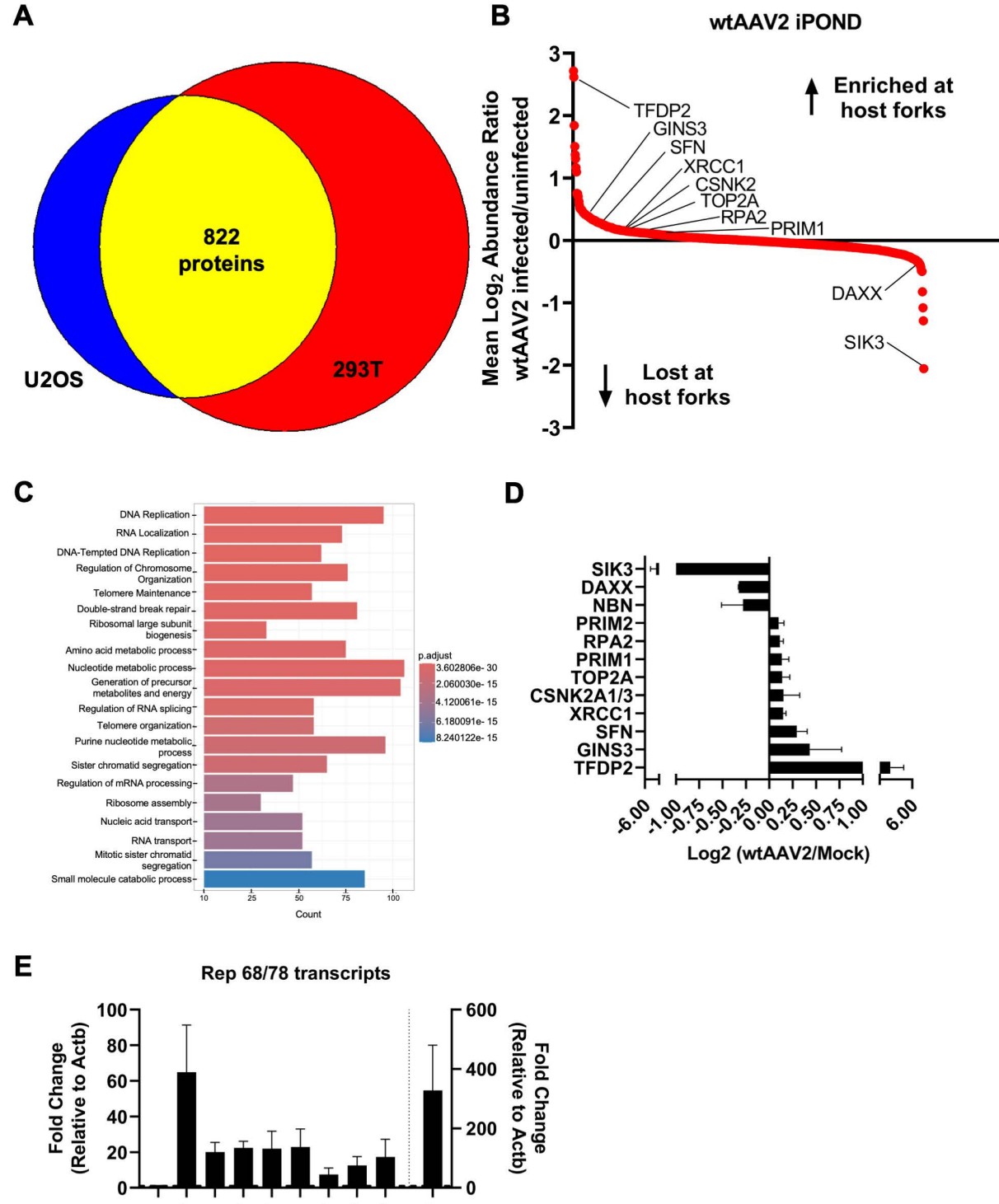

**Fig 3. Characterization of the wtAAV2-induced alterations to the human replisome using iPOND-MS.** **(A)** Venn diagram depicting the number of detected proteins enriched in 293T cells (red), U2OS cells (blue) and shared between the two cell types (yellow). (B) Scatter plot comparing the mean log2 abundance ratios between wtAAV2 infected and uninfected cells among the 822 proteins identified in the datasets in (A) to its corresponding rank

in decreasing order. (C) Gene ontology analysis of proteins enriched on nascent DNA in wtAAV2-infected 293T cells indicating the pathways that are associated with these proteins. (D) Comparison of protein enrichment on nascent DNA between Mock infected and wtAAV2-infected cells represented as mean of log2 values between U2OS and 293T cells focused on proteins indicated in Fig 3B. (E) RT-qPCR analysis of the Rep 67/78 transcript levels in wtAAV2-infected 293T cells at 24 hours post-infection that were transfected with the indicated siRNAs. The wtAAV2 transcripts are compared with Beta-actin transcripts as loading control and siMock-transfected cells set to one to calculate fold-change. Data is represented as mean±SEM of from two independent replicates of RNAi knockdowns followed by infections. Right y axis denotes the fold change relative to beta-actin in siCSKN1 transfected cells.

cellular replication stress monitored by DFAs and EdU-labelling, additionally identifying host factors that might regulated changes at the cellular replisomes.

## wtAAV2 genomes associate with cellular RPA and RPA overexpression rescues wtAAV2-induced replication stress

We have previously discovered that genomes of the related autonomous parvovirus MVM induces cellular DDRs by depleting the host genome of RPA2 molecules to exacerbate replication stress [23]. However, iPOND-MS analysis revealed an enrichment of RPA2 on the eukaryotic replisomes in wtAAV2-infected cells, presenting a conundrum. We hypothesized that a subset of RPA molecules are depleted by non-replicative wtAAV2 genomes, leaving a distinct population available to respond to stress on the host replisome. To determine whether non-replicative wtAAV2 genomes associate with components of the heterotrimeric RPA complex (made up of RPA14, RPA32 and RPA70 [63]), we performed ChIP-qPCR assays for these individual RPA subunits on the 5' end of the wtAAV2 genome. Interestingly, all three of the RPA subunits were bound to wtAAV2 genomes in the nucleus at 24 hpi (Fig 4A). To examine whether assembly of the entire RPA heterotrimer on the wtAAV2 genome leads to RPA activation, and if this depends on the number of wtAAV2 genome copies, we performed ChIP-qPCR for RPA32 phosphorylated at Serine 8 (which regulates the activation of replication checkpoint signals [64]) at increasing MOI's of wtAAV2 genomes in 293T cells at 24 hpi (Fig 4B). We discovered that high MOIs of wtAAV2 in 293T cells associated with phospho-RPA32, with a stepwise increase in interactions from 10,000 vg/cell to 20,000 vg/cell (Fig 4B). We confirmed that RPA32 phosphorylation is inhibited on the wtAAV2 genome using TDRL-505 [65], an allosteric inhibitor of RPA32's phosphorylation that blocks RPA32's DNA binding domain (Fig 4C). While inhibition of RPA32 phosphorylation by TDRL-505 (using the schematic illustrated in Fig 4D) was sufficient to shorten cellular replication forks monitored by IdU and CldU lengths (iRPA samples in Fig 4E,F), these defects on the host replication fork was not enhanced further in the presence of wtAAV2 infection (iRPA+wtAAV2 in Fig 4E,F). This suggests that the ability of RPA32 to bind the wtAAV2 genome or host replication forks plays a key role in protecting the genome from replication stress. Additionally, there might be redundant single-stranded DNA repair pathways that are activated upon RPA inhibition that synergistic induction of wtAAV2-induced replication stress. Strikingly though, inhibition of RPA32 activation led to derepression of the wtAAV2 genome, increasing the levels of Rep 68/78 transcripts generated in the 293T cells (Fig 4G, left) and U2OS cells (Fig4G, right). RPA32 inhibition also de-repressed the expression of a GFP transgene from rAAV2-transduced 293T cells (Fig 4H). These findings suggested that RPA32-mediated regulation of wtAAV2 gene expression is distinct from that of rAAV2. To overcome the impact of RPA-exhaustion induced by wtAAV2 genomes, we transfected plasmids expressing the RPA subunits (pRPA) into 293T cells prior to AAV2 infection as schematized in Fig 4D. Measurement of DNA fibers of these cells after wtAAV2 infection for 24 hours showed that pRPA transfection rescued wtAAV2-induced replication stress, elongating host replication fibers to higher than wild-type levels (IdU, Fig 4I) or approximately equal to wild-type lengths (CldU, Fig 4J).

## wtAAV2-mediated RPA32 exhaustion induces cellular DNA damage

To determine how competition for RPA32 molecules between host and the virus genomes (which we defined as wtAAV2-induced RPA exhaustion) impact cellular replication forks, we infected 293T cells for 24 hours with increasing the viral

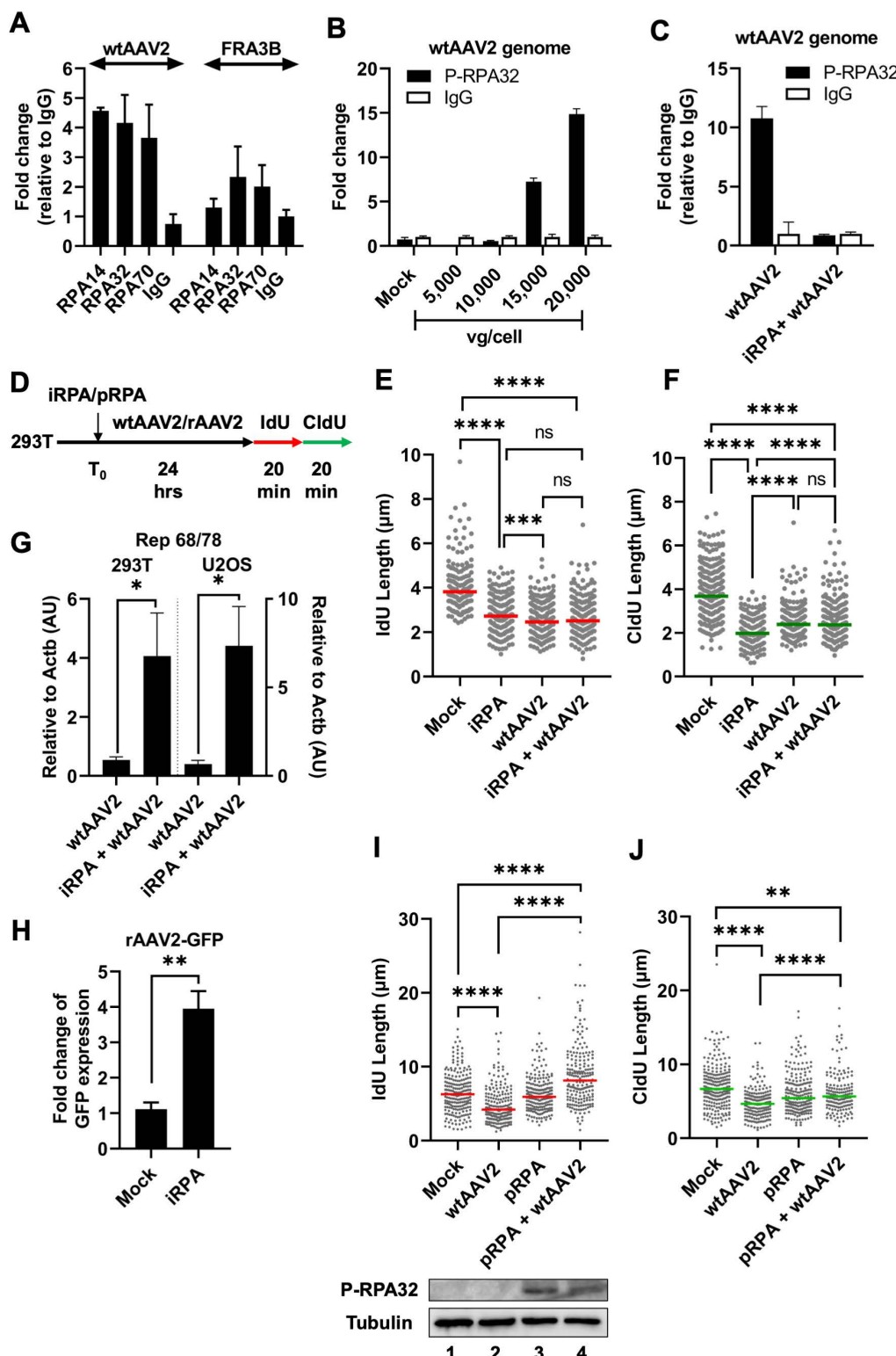

**Fig 4. wtAAV2 genomes associate with cellular RPA and RPA overexpression rescues AAV2-induced replication stress.** (A) Fold change of the binding of the indicated RPA subunits compared with isotype IgG to the wtAAV2 genome in 293T cells infected with wtAAV2 at an MOI of 10,000 vg/cell. Data is represented as mean±SEM of fold change. (B) ChIP-qPCR analysis of phosphorylated RPA 32 binding to the wtAAV2 genome at 24

hours post-infection at the indicated MOIs compared with IgG control. (C) Impact of RPA inhibition with TDRL on phosphorylated RPA binding to the wtAAV2 genome at 24 hpi in 293T cells infected at an MOI of 10,000 vg/cell. Background levels of the pulldown were evaluated using IgG as control. (D) Schematic of the timeline of wtAAV2 infection at an MOI of 10,000 vg/cell upon being pulsed with the RPA inhibitor TDRL or pRPA transfection in 293T cells for 24 hours before sequential pulsing with IdU and CldU for 20 minutes each prior to DNA fiber analysis. (E, F) DNA fiber length measurements were indicated by each datapoint for (E) IdU-labelled DNA fibers and (F) CldU-labelled DNA fibers. The median lengths of many measurements of IdU and CldU labelled fibers in wtAAV2 infected cells are represented by red bar and green bar respectively. At least 150 individual fibers were measured for each condition for three independent replicates of wtAAV2 infections of 293T cells. Statistical significance was measured by Mann Whitney Wilcoxon test, **** represents P<0.0001, ns represents differences that are not statistically significant. (G) wtAAV2 gene expression was monitored by RT-qPCR for Rep 68/78 transcripts compared with cellular *Actb* levels in AAV2 infections of 293T cells (left) and U2OS cells (right) in the presence of the RPA inhibitor TDRL-505 schematized in (D). Data presented as mean±SEM of Rep 68/78 transcripts relative to *Actb* from at least three independent experiments. (H) Quantification of GFP expression levels relative to beta actin in 293T cells treated with TDRL-505 prior to rAAV-GFP transduction according to the schematic in Fig 4D. Data is represented as fold change of GFP versus beta Actin relative to Mock-treated cells. DNA fiber length measurements indicated by each datapoint for (I) IdU-labelled DNA fibers and (J) CldU-labelled DNA fibers in AAV2 infected cells in the presence of transfected pUC18/pRPA represented by red bar and green bar respectively as schematized in 4D. The bottom half of I shows the expression of pRPA30 in transfected cells (top) relative to tubulin loading control (bottom). At least 150 individual fibers were measured for each condition for three independent replicates of wtAAV2 infections of 293T cells. Statistical significance was measured by Mann Whitney Wilcoxon test, **** represents P<0.0001, ns represents differences that are not statistically significant.

genomes per cell before performing DFAs by sequential IdU/CldU pulses (schematized in Fig 5A). As shown in Fig 5B for IdU lengths and Fig 5C for CldU lengths, increase of MOIs of wtAAV2 genomes infecting 293T cells led to a dose-dependent decrease in cellular DNA synthesis. To determine whether high levels of replication stress was associated with cellular DNA damage, we labelled wtAAV2-infected U2OS cells (at a high MOI of 10,000 vg/cell at 24 hours) with EdU for 20 minutes, co-stained for γH2AX before performing confocal imaging. EdU labelled sites in mock-infected cells did not associate with γH2AX foci (Fig 5D, top panel). Strikingly however, in AAV2-infected cells, the EdU labelled sites (green) colocalized with γH2AX signals (red; Fig 5D, bottom). To determine whether induction of dose-dependent replication stress by wtAAV2 leads to cellular DNA damage, we measured the γH2AX foci in EdU-labelled U2OS cells infected with wtAAV2 at different MOIs. As shown in Fig 5E, increasing the wtAAV2 genomes per cell led to an increase of γH2AX foci per cell in EdU-positive cells. To confirm the dose-dependent induction of cellular DNA damage signals, we performed western blots for γH2AX in 293T cells infected with wtAAV2 at progressively increasing doses, showing that γH2AX levels were enhanced in cells infected with wtAAV2 at MOI's of 10,000, 15,000 and 20,000 viral genomes per cell (Fig 5F, 3-5 Lanes). We validated these observations in Normal Human Dermal Fibroblasts (NHDF's) that have low basal levels of γH2AX (Fig 5G, 1 Lane), discovering that wtAAV2 genomes at high doses induce cellular γH2AX (Fig 5G, 3 Lane). We confirmed the generalizability of wtAAV2-induced DNA damage in independent cell lines including A549 (S4A Fig), HepG2 (S4B Fig); as well as the ability of rAAV2 (S4C Fig) and scAAV2 (S4D Fig) to induce cellular DNA damage in 293T cells. To determine where these wtAAV2-induced DDR signals colocalize in the nucleus the viral genome, we performed Immuno-FISH imaging of wtAAV2 genomes (green) with the phosphorylated versions of RPA32, NBN (also known as NBS1) and MDC1. As shown in Fig 5H, all the phosphorylated versions of these DDR proteins colocalized with wtAAV2 genomes in the nucleus of infected U2OS cells. Surprisingly, both rAAV2 and scAAV2 genomes occupied nuclear spaces that were distinct from that of phosphorylated cellular DDR markers (S5A and S5B Fig respectively). These findings suggested that the increase of wtAAV2 genomes in host-cell nuclei generate a shortening of cellular replication forks by inducing RPA exhaustion that can lead to DNA damage on the cellular genome in the vicinity of wtAAV2.

## Discussion

In this study we have discovered that wtAAV2 mono-infection as well as rAAV2/scAAV2 transduction at high MOI induces replication stress on the host genome. Since rAAV2/scAAV2 are sufficient to induce replication stress, despite lacking the Rep 68/78 ORFs, these observations suggest that the wtAAV2 Rep 68/78 proteins are not necessary to induce cellular replication stress. We also observed that neither the ITR element nor empty capsids induced host-cell replication stress. We interpret these results to indicate that the incoming unreplicated single-stranded wtAAV2 genome is driving replication

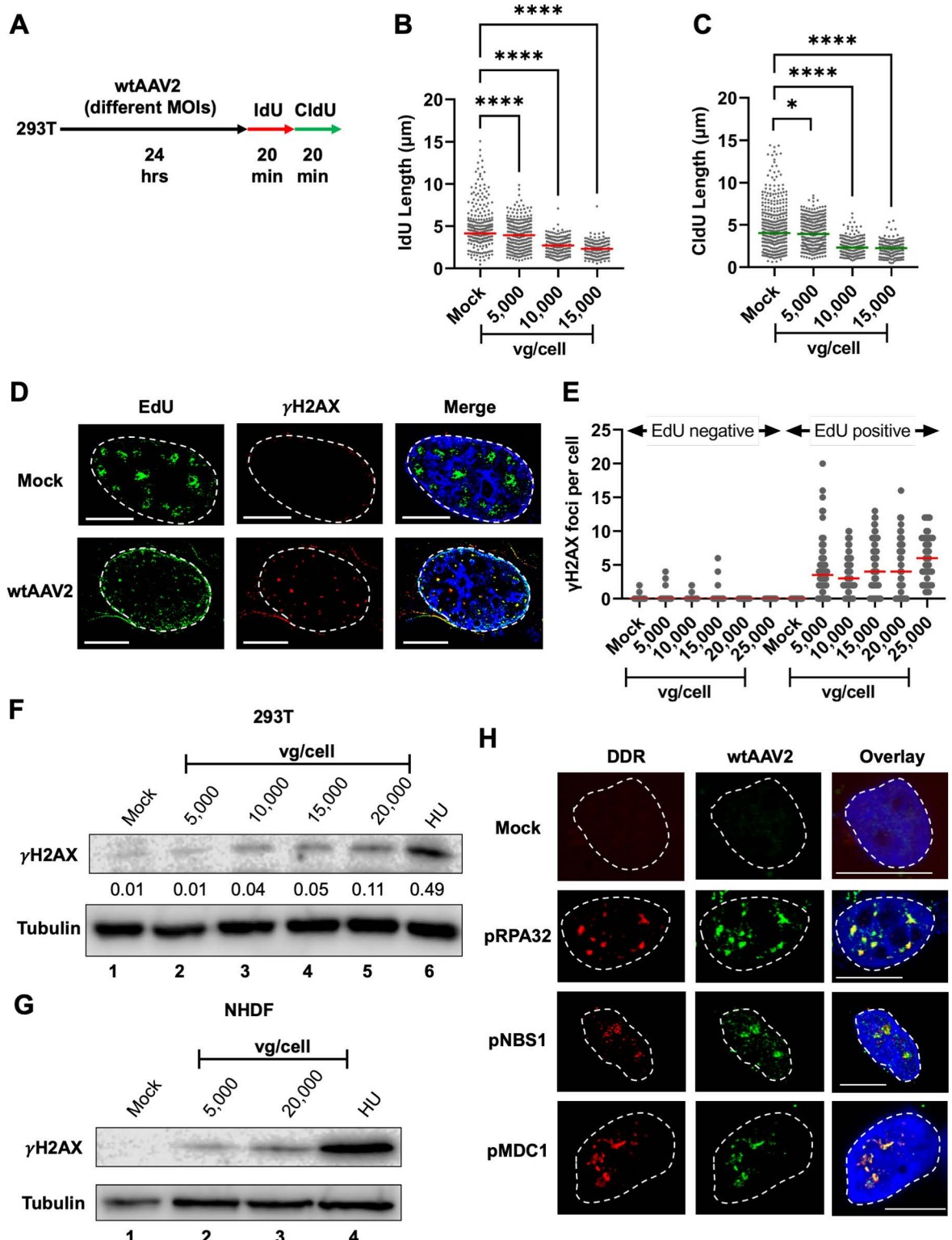

**Fig 5. wtAAV2-mediated RPA32 exhaustion induces cellular DNA damage.** (A) Schematic of the timeline of wtAAV2 infection at different multiplicities in 293T cells for 24 hours before sequential pulsing with IdU and CldU for 20 minutes each prior to DNA fiber analysis. (B,C) DNA fiber length measurements were indicated by each datapoint for (B) IdU-labelled DNA fibers and (C) CldU-labelled DNA fibers. The median lengths of many

measurements of IdU and CldU labelled fibers in wtAAV2 infected cells are represented by red bar and green bar respectively. At least 150 individual fibers were measured for each condition for three independent replicates of wtAAV2 infections of 293T cells. Statistical significance was measured by Mann Whitney Wilcoxon test, **** represents P<0.0001, ns represents differences that are not statistically significant. (D,E) U2OS cells infected with wtAAV2 at an MOI of 10,000 vg/cell were pulsed with EdU (green) for 20 minutes before being processed for gamma H2AX localization (red). The nucleus is shown by DAPI staining (blue) with the nuclear border demarcated by dashed white line. (E) Quantification of the number of gamma H2AX foci per cell in U2OS cells that were EdU-negative (left) and EdU-positive (right) upon being infected with wtAAV2 at different MOI's for 24 hours. The median number of gamma H2AX foci in at least 30 nuclei is indicated by red bar. (F,G) Immunoblot analysis of AAV2 infection at the indicated MOI's compared with 2 mM HU for 24 hours in (F) 293T cells and (G) NHDF cells. Cells were harvested as described and analyzed for γH2AX in the nuclear lysates. Tubulin levels were used as loading controls for the immunoblots. (H) Immuno-FISH assays imaging the relative location of the wtAAV2 genome (green) to active forms of the identified cellular DDR factors (red) in iPOND-MS studies described above. U2OS cells were infected with wtAAV2 at an MOI of 10,000 vg/cell for 24 hours before being processed for Immuno-FISH imaging. Nuclei are represented by blue DAPI staining and the nuclear borders are demarcated by dashed white lines. The scale bars represent 10 microns distance.

stress in the infected cells. The observation that wtAAV2 genomes induce replication stress within to 12–24 hours post-infection suggests that acute stress-response signals are likely not involved in shortening cellular DNA synthesis. This is critical because replication stress has recently been connected to innate immune activation upon reactivation of endogenous retroviruses [66]. The absence of further decrease of DNA synthesis at 36 hpi is distinct from that observed in replicating autonomous parvoviruses like MVM where there continues to be a decrease in DNA fiber lengths till cell death ([23]; Fig 6). An alternate explanation for this outcome might be that cells that have undergone division within this window have diluted the number of wtAAV2 genomes inside each cell nucleus, leading to a rescue of RPA exhaustion-mediate stress induction. As a corollary, host-cell RPA molecules associate with wtAAV2 genomes at high-doses, likely rendering the cellular genome vulnerable to replication stress. Exacerbated by large number of wtAAV2 genomes in the nucleus, RPA exhaustion causes shortening of cellular replication forks that leads to the induction of cellular DDRs.

Our findings of non-replicative wtAAV2 genomes inducing replication stress mirror the observations made with UV-inactivated MVM genomes that cause replication stress via RPA exhaustion [23]. Importantly however, while UV-inactivated MVM genomes did not induce cellular γH2AX signals [1], our findings show that wtAAV2 at high multiplicity did lead to induction of cellular γH2AX. These cellular DDR signals might be generated by virally induced challenges to cell cycle progression. MVM infection inhibits the DREAM complex protein FoxM1 to inhibit transcription of the mitotic entry protein *Ccnb1*, leading to a potent pre-mitotic cell cycle block at the G2/M border [67]. On the other hand, wtAAV2 infection enriches the transcription factor TFDP2, which interacts with E2F proteins to regulate G1/S checkpoint entry. However, since RNAi-mediated TFDP2 knockdown did not impact wtAAV2 Rep 68/78 transcripts, our studies suggests that factors like TFDP2 regulate proteins on the host genome and not on the viral genome. While it is important to note that wtAAV2-induced replication stress does not arrest wtAAV2-infected cells in S phase, as is the case for parvoviruses like HBoV [68], this replication stress leads to the induction of cellular DNA damage at high multiplicities. These observations suggest that cellular response pathways are likely activated to overcome wtAAV2-induced replication stress at low multiplicities, allowing for faithful host DNA replication to progress. However, the identity of these host response pathway proteins and how they function remain unknown. We hypothesize that targets identified in our iPOND analysis, such as GINS3, TOP2A and XRCC1, may play a role in maintaining host replisome stability as wtAAV2 mono-infection progresses long-term. It would be important to assess the role of these factors in long-term wtAAV2 infection in primary cells or non-virally transformed models of rAAV2 transduction.

Recent investigations of iPOND-MS in viral systems have extensively profiled the host factors that are usurped by viruses to amplify their genomes, including polyomaviruses [69], herpesvirses [70–72] and adenoviruses [70,73]. Since wtAAV2-mono-infection does not generate nascent DNA at late timepoints, our studies are the first to profile the impact of viral infection exclusively on the host replisome. Indeed, our iPOND-MS analysis point to two distinct possibilities by which viral infection might cause replication stress: 1) formation of protein-DNA complexes, such as those formed by transcription-replication collisions on the host genome; or 2) depletion of replication stress proteins at eukaryotic

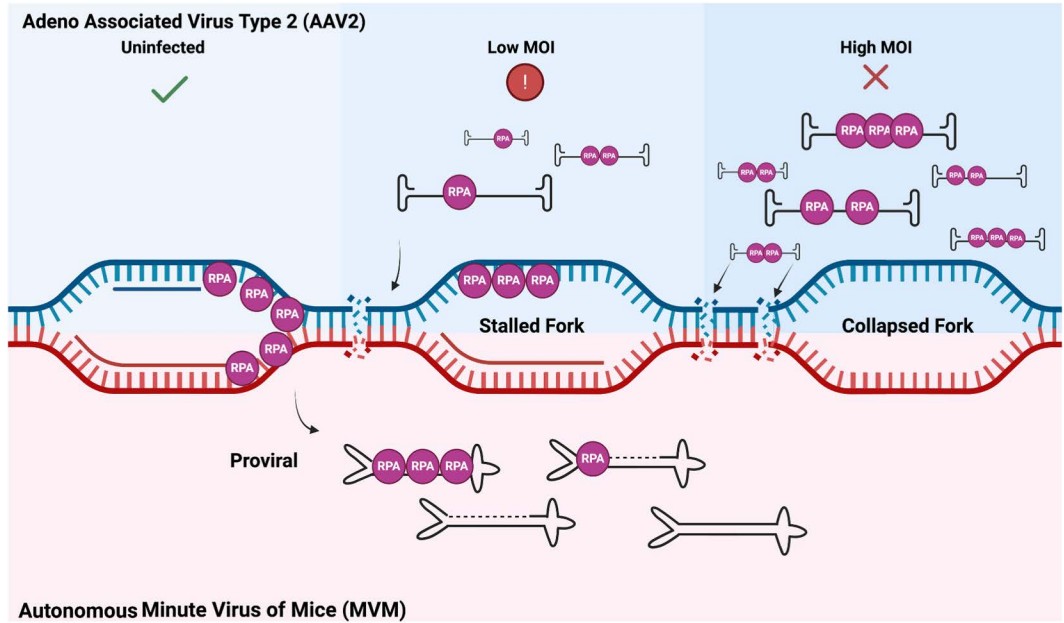

**Fig 6. Mechanism of wtAAV2-induced RPA exhaustion that leads to replication stress on the host genome.** wtAAV2 infection depletes the host genome of RPA molecules, rendering the host replisomes vulnerable to replication stress (middle panel). At high MOI's of wtAAV2 infection, this RPA exhaustion leads to induction of cellular DNA damage (right panel) while restricting viral gene expression. Replication of the autonomous parvovirus MVM, however, leads to induction of a cellular DDR via RPA exhaustion that has pro-viral effects. Image generated using Biorender.

replisomes by competition from wtAAV2 genomes. In support of both possibilities, iPOND-MS revealed the enrichment of RNA processing machinery and proteins that regulate replication checkpoints at nascent DNA in wtAAV2-infected cells. Indeed, the enrichment of PARP1 and CSNK1A2 (Casein Kinase subunits) have previously been found to associate with Rep 68/78 [74] and assembled capsids [75] respectively. CSNK1A2 has been proposed to serve as a host restriction factor that regulates rAAV2 transduction [62]. Additionally, during MVM infection, the viral non-structural protein NS1 redirects Casein Kinase subunits to suppress local ATR signaling [40]. The details of how ATR-mediated signals may synergize with Casein Kinase to regulate wtAAV2 life cycle and reduced host DNA synthesis warrants further investigation.

Our iPOND, RNAi and inhibitor studies suggest that RPA plays a key role in wtAAV2 life cycle and wtAAV2-induced replication stress. The association of all three RPA subunits (RPA14, RPA32 and RPA70) on the wtAAV2 genome is striking because RPA canonically binds single-stranded DNA but by 24 hpi most of the wtAAV2 genomes should have been converted into its double stranded form [76,77]. It is noteworthy, however, that RPA does bind to double-stranded DNA, albeit at lower binding strength than single-stranded DNA [78]. Alternatively, it is possible that RPA subunits associate with wtAAV2 genome components that are partially accessible, such as the viral replication origin, polypyrimidine tracks in the ITR or viral promoters. The ChIP-qPCR assays deployed for these studies only sonicate the wtAAV2 genome to sizes of 500 base pairs or less, making it difficult to resolve the position of the RPA molecules accurately. Regardless, the derepression of wtAAV2/rAAV2 gene expression upon RPA inhibition indicates that the RPA subunits can directly or indirectly block wtAAV2/rAAV2 gene expression. One possibility is that RPA directly prevents the access of transcription factors by competitive inhibition. Alternatively, RPA might recruit host factors that transcriptionally silence wtAAV2/rAAV2 gene expression. An example of this possibility is RPA-mediated recruitment of chromatin modifying enzymes such as G9A [79], a histone methyltransferase that is known to methylate host chromatin [80]. These observations are parallel to the binding of KAP1 to the wtAAV2 genomes that recruit the chromatin modifier SETDB1 leading to wtAAV2 gene repression

[81]. Future studies will build on how DDR factors on the wtAAV2 genome generally regulate the chromatin landscape of wtAAV2 and how this regulates the non-replicative life cycle of wtAAV2.

Immuno-FISH imaging of wtAAV2 genomes colocalizing with replication and DDR factors corroborate prior findings of wtAAV2's interaction with DDR sites [27]. These nuclear foci are likely sites where the wtAAV2 genomes undergo processing to be converted from single- to double-stranded DNA molecules, perhaps forming the stalled replication fork-like structures that have previously been described [41]. Since wtAAV2 infection induces replication stress that can lead to DNA damage, we surmise that wtAAV2-induced replication stress is focused on cellular sites that are in proximity to the viral genome by usurping RPA molecules from the host. We propose that this competition of RPA molecules renders the host genome vulnerable to replication stress that can lead to DSBs (Fig 6). These observations are analogous to MVM genome-induced RPA exhaustion that we have previously discovered ( [23], Fig 6). However, unlike replicating MVM genomes, non-replicative wtAAV2 genomes do not generate multiple single-stranded and double stranded intermediates, rendering the possibility of RPA binding to nascent genome structures unlikely. Moreover, while RPA exhaustion seems to induce more cellular DNA damage leading to a pro-viral effect on MVM life cycle, these functions have a repressive effect on wtAAV2 (Fig 6). Taken together, our studies reveal the molecular underpinnings of how wtAAV2-induced replication stress can cause cellular DNA damage.

## Materials and methods

### Cell lines

Female human U2OS osteosarcoma cells, female human embryonic kidney cells (HEK293T) and male human alveolar basal epithelial (A549) cells were cultured in Dulbecco's modified Eagle's medium (DMEM, high glucose; Gibco) supplemented with 5% Serum Plus (Sigma Aldrich) and 50 µg/ml gentamicin (Gibco). Male hepatocellular carcinoma cells (HepG2) were cultured in DMEM F12 (Gibco), 5% Fetal Bovine Serum (FBS) and 1% Penicillin Streptomycin. 200,000 HepG2 cells were cultured onto 100mm dishes coated with Rat Tail Collagen for 30 minutes. Expi293T cells were cultured in Expi293 Expression Medium (Gibco) and were used for AAV2 virus production from the SSV9 infectious clone. Normal Human Dermal Fibroblasts (NHDFs) were obtained from Dr. Robert Kalejta and were cultured in DMEM supplemented with 5% FBS. Cells were cultured in incubators in 5% CO2 at 37 degrees Celsius. Cell lines are tested for mycoplasma contamination and background levels of DNA damage by γH2AX staining.

### Virus and viral infection

wtAAV2 virus was produced in Expi293T cells and HEK293T cells as described previously [1]. wtAAV2, scAAV2, and rAAV2 infection and transduction was carried out at a Multiplicity of Transfection (MOT) of 5,000 viral genomes/cell unless otherwise noted. Empty wtAAV2 capsids were obtained from PROGEN (American Research Products Inc.).

### RNAi, transfections and inhibitors

The RNAi sequences (Life Technologies) to silence the respective genes are: RPA2 (GCACCUUCUCAAGCCGAAAtt), TOP2A (*GGAUUCUGCUAGUCCACGAtt*), PRIM1 (*GAACCAGAGAUUAUAAGAAtt*), SIK3 (*CAGCGACGAUGCUUAU-GUAtt*), TFDP2 (*GGACUACUUCUGAACUCUAtt*), GINS3 (*GACUUUCAGUGUUGGGAGAtt*), SFN (*ACUUUUCCGU-CUUCCACUAtt*), XRCC1 (*GGCAGACACUUACCGAAAAtt*) and CSNK2A1 (*GGCUCGAAUGGGUUCAUCUtt*). RNAi constructs were transfected using Lipofectamine RNAiMAX Transfection kit (Life Technologies) into 293T cells for 24 hours. The RPA inhibitor, TDRL-505 (Millipore), was used at a final concentration of 50 µM in 293T and U2OS cells. For AAV2 virus production, Expi293T cells were transfected with the ExpiFectamine 293 Transfection Kit (Thermo Scientific). The non-palindromic core ITR sequence used to measure ITR-induced replication stress was: ggccgcccgggcaaagcccggg-cgtcgggcgacctttggtcgcccggcc (IDT). The RPA over-expression plasmid (pRPA) has previously been published [82] and 0.5

µg was transfected with Lipo293D transfection reagent (SignaGen). An equivalent amount of pUC18 plasmid was used as mock transfection control.

## DNA fiber assay (DFA)

HEK293T or U2OS cells were transduced or infected according to experimental requirements (as described above and previously [83]). Single molecule DNA Fiber Analysis experiments were completed following previously published protocols [58]. Cells were pulsed in 20mM IdU for 20 minutes at the end of infection, followed immediately by pulsing with 50mM CldU for 20 minutes. Cells were pelleted at 5000xg for 5 minutes and resuspended in 200 µL of complete media, then stored on ice for the duration of the protocol. 2 µL of resuspended cell solution was pipetted onto positively charged slides, then mixed with 6 µL of DFA Lysis Buffer (200 mM Tris-HCl pH 7.5, 50 mM EDTA, 0.5% SDS) and allowed to lyse for 5 minutes. Slides were then tilted to spread the DNA fibers along the slide and air dried for 15 minutes. DNA was fixed using a 3:1 methanol:acetic acid solution. The DNA was denatured using 2.5 M HCl for 1 hour at room temperature. After denaturing, slides were blocked in 3% BSA in PBS for 30 minutes in a humidified chamber at room temperature. Primary antibody staining was carried out using Abcam rat anti-BrdU (1:1000) and BD Biosciences mouse anti-BrdU (1:500) at room temperature for 30 minutes, then washed with 0.1% Tween 20 in PBS 3 times. Slides were then stained with anti-rat Alexa Fluor 488 and anti-mouse IgG1 Alexa Fluor 568 (1:1000) secondary antibodies at room temperature for 30 minutes in the dark. Samples were washed with 0.1% Tween 20 in PBS 3 times and cover slips were affixed to slides using Pro-Long Gold Antifade Mountant (Thermo Scientific). Fibers were imaged with a Leica Stellaris confocal microscope using a 63X oil immersion objective lens. Fiber lengths were measured using Digimizer software (MedCalc Software Ltd).

## Antibodies

Antibodies used for DNA fiber analysis were: anti-BrdU (BD Biosciences, Clone B44, 347580), anti-BrdU (Abcam, ab6326), Alexa-Fluor 568 conjugated anti-mouse secondary (Thermo Scientific, A11004), Alexa-Fluor 488 conjugated anti-rat secondary (Thermo Scientific, A11006).

Antibodies used for western blot: γH2AX (AbCAM, ab11174), Tubulin (Millipore, 05-829), REP68/78 (IF11.8); ChIP-qPCR analysis: RPA14 (Life Technologies, MA1-23281), RPA32 (Cell Signaling, 52448S), RPA70 (Cell Signaling, 2267S), phospho-RPA32 Ser 8 (Cell Signaling, 83745), TFDP2 (Protein Tech, 11500-1-AP), PRIM1 (Protein Tech, 10773-1-AP), XRCC1 (Cell Signaling, 20026T) and SIK3 (Thermo Scientific, PAS-115813); Immuno-FISH: phospho-MDC (AbCAM, ab36513) and phospho-NBS1 (Cell Signaling, 3001).

## Chromatin immunoprecipitation combined with quantitative PCR (ChIP-qPCR)

HEK293T cells were infected with wtAAV2 as described above and were crosslinked in 1% Formaldehyde for 10 minutes at room temperature. 0.125 M glycine was used to quench the crosslinking reaction for 5 minutes at room temperature. Cells were lysed on ice for 20 minutes in ChIP lysis buffer (1% SDS, 10 mM EDTA, 50 mM Tris-HCl, pH 8, protease inhibitor). Cell lysates were sonicated using a Diagenode Bioruptor Pico for 60 cycles (30 s on and 30 s off per cycle), before being incubated overnight at 4 degrees C with the antibodies bound to Protein A Dynabeads (Invitrogen). Using low-salt wash (0.01% SDS, 1% Triton X-100, 2 mM EDTA, 20 mM Tris-HCl pH8, 150 mM NaCl), high salt wash (0.01% SDS, 1% Triton X-100, 2 mM EDTA 20 mM Tris-HCl pH8, 500 mM NaCl), lithium chloride wash (0.25M LiCl, 1% NP40, 1% DOC, 1 mM EDTA, 10 mM Tris HCl pH8) and twice with TE buffer samples were washed for 3 minutes each at 4 degrees. Using SDS elution buffer (1% SDS, 0.1M sodium bicarbonate) DNA was eluted and crosslinks were reversed using 0.2M NaCl, Proteinase K (NEB) and incubated at 56 degrees C overnight. Using a PCR Purification Kit (Qiagen) DNA was purified and eluted in 100 µl of Buffer EB (Qiagen). ChIP DNA was quantified by qPCR analysis (Biorad) under the following conditions: 95° C for 5 mins, 95° C for 10 secs and 60° C for 30 secs for 40 cycles. wtAAV2 genome interaction with RPA

molecules was assessed by qPCR assays using primers complementary to the REP open reading frame. These values were compared with primers monitoring the levels of Beta actin transcripts. The primer sequences used for RT-qPCR and ChIP-qPCR on the Rep 68/78 ORF in 5' to 3' orientation are: TGATAAGCGGTTCAGGGAGT (forward primer) and CCAG-CCATGGTTAGTTGGTT (reverse primer); human genome on the FRA3B gene: CCCCAAACTGTCCCAGTAGA (forward primer) and TGGATCCTGGCAGAGACTTT (reverse primer); GFP gene: ACGTAAACGGCCACAAGTTC (forward primer) and AAGTCGTGCTGCTTCATGTG (reverse primer); Beta actin gene: CACCTTGATCTTCATTGTGCTG (forward primer) and GCAAAGACCTGTACGCCAAC (reverse primer).

## Statistical analysis

Statistical analysis was performed using Graphpad Prism software and appropriate statistical test as indicated in the manuscript.

## iPOND-SILAC-MS

iPOND was performed as previously described [61]. Cells (U2OS or 293T) were labeled with 10 µM EdU for 10 minutes following 24-hour infection at an MOI of 5,000 vg/cell with wtAAV2 virus. Cells were cross-linked in 1% fresh formalde-hyde/PBS for 10 minutes (RT), quenched with 1.25 M glycine, and washed with PBS. Collected cell pellets were frozen at -80 C. The next day pellets were permeabilized in 0.25% Triton-X/PBS and washed once with 0.5% BSA/PBS followed by PBS. Light and heavy labeled cells were mixed 1:1 by cell number and the click reaction was completed in 1 hour with PEG4-biotin azide (Invitrogen). The cells were subsequently lysed by sonication using a Diagneode Pico sonicator on ultra-high setting until homogenized. DNA-protein complexes were purified using streptavidin-coupled C1 magnetic beads for 1 hr. Samples were washed (5 minutes each) with lysis buffer (1% SDS in 50 mM Tris, pH 8.0), low salt buffer (1% Triton X-100, 20 mM Tris, pH 8.0, 2 mM EDTA, pH 8.0, 150 mM NaCl), high salt buffer(1% Triton X-100, 20 mM Tris, pH 8.0, 2 mM EDTA, pH 8.0, 500 mM NaCl), lithium chloride buffer (100 mM Tris, pH 8.0, 500 mM LiCl, 1% Igepal), followed by two washes in lysis buffer. Captured proteins were eluted and cross-links were reversed in 6X SDS sample for 30 min at 95 C while mixing every 10 minutes.

All incubations were carried out at room temperature unless otherwise specified. iPOND eluates were separated on 4–12% Tris-glycine mini gels (Novex, Invitrogen). Following electrophoresis, gels were incubated for 30 min in fixing solution (50% methanol, 1.2% phosphoric acid) to immobilize proteins. After a brief rinse in water, gels were stained overnight in a solution containing 50% methanol, 1.2% phosphoric acid, 1.3 M ammonium sulfate, and 0.1% (w/v) Coomassie Brilliant Blue G-250.

Each lane (sample) was then cut into small rectangular pieces and divided into two fractions: fraction 2 (containing the streptavidin band) and fraction 1 (containing the remaining proteins). For in-gel digestion, both fractions were subjected to several rounds of washing to remove salts and Coomassie. Specifically, gel pieces were incubated for 15 min in 25 mM ammonium bicarbonate, 50% ethanol (60 °C for strongly stained pieces), and the supernatant was discarded. This wash step was repeated 2–3 times or until gel pieces were destained completely. Next, in preparation for cysteine reduction & alkylation, fractions were incubated in 100% ethanol for 10 min to dehydrate the gel pieces.

Cysteine residues were reduced by incubating fractions for 20 min at room temperature in 20 mM DTT, 50 mM ammonium bicarbonate. After discarding the supernatant, cysteine residues were alkylated by a 20-min incubation in 80 mM chloroacetamide, 50 mM ammonium bicarbonate. Alkylation compounds were removed with successive 10-min washes in i) 50 mM ammonium bicarbonate, ii) 25 mM ammonium bicarbonate, 50% ethanol, and iii) 100% ethanol. Gel pieces were subsequently dried in a vacuum concentrator.

For overnight digestion at 37 °C, trypsin solution (0.25 µg trypsin [Promega] in 50 µl of 50 mM ammonium bicarbonate) was added to each fraction. On the following day, peptides were extracted by two sequential 15-min elutions in 150 µl of elution solution (80% acetonitrile, 0.2% formic acid) under sonication. The resulting eluates were pooled and evaporated to dryness in a vacuum concentrator.

Prior to LC–MS analysis, dried peptide samples were subjected to a final cleanup using solid-phase extraction (SPE) cartridges (1 cc, 50 mg C18 sorbent, Sep-Pak, Waters) in conjunction with a vacuum manifold. Samples were first resuspended in 200 µl of 0.2% formic acid. The cartridges were then equilibrated with 100% acetonitrile, followed by two washes with 0.2% formic acid. After loading the samples onto the cartridges, two additional washes with 0.2% formic acid were performed. Finally, peptides were eluted with 80% acetonitrile, 0.2% formic acid, and the eluates were dried to completion in a vacuum concentrator. Prior to LC–MS analysis, dried peptide samples were resuspended in 0.2% formic acid and their concentrations were determined using a NanoDrop spectrophotometer.

## LC-MS analysis

For LC–MS analysis, a Vanquish Neo UHPLC system (Thermo Fisher Scientific) was coupled to an Orbitrap Ascend mass spectrometer (Thermo Fisher Scientific) via a Nanospray Flex ionization source. A 40 cm in-house–prepared fused silica C18 column was maintained at 50 °C using a custom-built column heater. The source voltage was set to 2 kV, and the ion transfer tube was held at 275 °C. Peptide samples from streptavidin-containing fractions were injected at a total amount of 500 ng, whereas 1 µg of peptides was loaded for all other samples. Peptides were separated at a flow rate of 300 nL/min with a 73 min active gradient from 5% to 46% solvent B (solvent A: 0.2% FA; solvent B: 80% acetonitrile with 0.2% FA).

Orbitrap Ascend parameters were as follows: MS1 scans were acquired at a resolution of 240k (scan range 300–1350 m/z), with a maximum injection time of 50 ms, an automatic gain control (AGC) target of $1 \times 10^6$, a normalized AGC target of 250%, and an RF lens percentage of 30. Data-dependent acquisition employed a 1 s cycle time. MIPS (monoisotopic peak determination) was set to "peptide", and the isolation window center was set to "most abundant peak". Charge states 2–5 were included, and dynamic exclusion was set to 20 s with a ± 5 ppm mass tolerance. MS2 spectra were recorded in the ion trap using an isolation window of 0.8 m/z, a normalized HCD collision energy of 24%, an ion trap scan rate set to "Turbo," a scan range of 150–1350 m/z, a normalized AGC target of 250%, and a maximum injection time of 12 ms.

## Proteomics data analysis

RAW files generated from the LC–MS runs were analyzed with MaxQuant (version 2.4.2.0), searching against a UniProt human FASTA file containing both SwissProt and TrEMBL entries. Adeno-associated virus 2 proteins (Rep68 [P03132] and Rep78 [Q89268]) were added to this database. Unless otherwise specified, all MaxQuant parameters were set to their default values.

Gel fractions originating from the same lane were assigned identical experiment names but treated as separate fractions in MaxQuant. In the *Group-specific parameters* tab, "Type" was set to *multiplicity = 2*, and *Arg10* and *Lys8* were checked. Under *Misc.*, *Re-quantify* was enabled. In the *Global parameters* tab, the FASTA files described above were specified under *Sequences*. Under *Protein quantification*, *Label min. ratio count* was set to *1*, and under *Identification*, *Min. unique peptides* was set to *1* with *Match between runs* enabled. Finally, under *Label-free quantification*, *iBAQ* was enabled.

R (v4.4.3) was used to visualize mass spectrometry–derived gene expression data, integrating gene ontology (GO) annotations and essential biological pathways. Data were imported with readr, and biomart annotations retrieved via biomartr. Gene IDs were mapped to GO terms and pathways using org.Hs.e.g.,db, followed by functional enrichment analysis with clusterProfiler. High-quality, customizable plots were generated in ggplot2, depicting enriched GO terms and pathway networks. Analyses were conducted in RStudio to ensure reproducibility and modular code.

## wtAAV2 virus production

Expi293F cells were passaged to a cell density of $2.0x10^6 – 3.0x10^6$ cells/mL resuspended in 25 mL of fresh Expi293F media and expected to double within 24 hours when incubated at 37 degrees C, 8% CO2 in a shaking incubator. To prepare for transfection, cell culture volumes were expanded to 1L at the same density, then diluted according to

manufacturer's specifications. PEI (1 mg/mL concentration) was used as the transfection method at a ratio of 3:1. Plasmid DNA (1 µg/mL of cell culture) and PEI were diluted in OPTI-MEM media in separate tubes according to manufacturer's guidelines. These mixtures were combined and incubated for 10–15 minutes at room temperature to generate transfection complexes. The complexed mixtures were added to the cell culture dropwise as the flask was being swirled. The next day of transfection, enhancer 1 and 2 solutions were added from the Expifectamine Transfection kit (Thermo Scientific) and placed back into the incubator for the remainder of the incubation. After 5–7 days of incubation the cells were collected into 50 mL falcon tubes and centrifuged for 2 minutes at maximum speed. The pellet was resuspended in 15 mL of TE buffer. A series of seven freeze-thaw cycles were performed by alternating between liquid nitrogen (for freezing) and $37^0$C incubator (for thawing). After the freeze/thaw cycles DNase I (Thermo Scientific) treatment was carried out with 0.5U/µL for 1 hour at $37^0$C. The cells were then centrifuged at 2,000xg at $4^0$C for 10 minutes and the supernatant was transferred into a screw cap tube and stored in $4^0$C for temporary storage, and $-80^0$C for long-term storage.

### Immuno-FISH imaging

DNA probe was generated by labelling oligonucleotides complementary to the wtAAV2 genome using Aminoallyl-1-dUTP and TdT (Thermo Scientific). The oligos were ethanol precipitated and NHS-ester dyes (Thermo Scientific) were conjugated using sodium bicarbonate. Probes were purified using PCR clean-up column (Promega) and dissolved in TE buffer, as previously described [84]. After 16–24 hours post infection, cells were pre-extracted in Cytoskeletal (CSK) buffer for 3 minutes followed by 1 mL of CSK+ Triton X solution for 3 minutes. CSK buffers were aspirated, cells washed with 1 mL PBS and fixed with 4% paraformaldehyde in PBS for 10 min at room temperature (RT). Cells were washed with 1 mL of PBS and permeabilized in 1 mL of Permeabilization Buffer for 10 minutes. 10% formamide in 2X SSC buffer (300 mM sodium chloride, 30 mM sodium citrate) was added to the cells for 1–2 hours to rehydrate and pre-denature the DNAs. The probes were suspended in hybridization buffer to a final concentration of 0.5 ng/µL. Samples were hybridized with the probe solution in 2XSSC and sealed on glass slide with rubber cement and hybridized overnight at 37 degrees C in a humidified chamber. Cover slips were washed in 2X SSC/0.1% Triton X-100 at 37 degrees for 3 minutes each, 2X SSC at 37 degrees C and mounted on slide with Prolong Diamond Antifade Mounting Media with DAPI. Samples were imaged on a confocal microscope (Leica) with 63X oil objective lens.

### EdU labeling coupled with immunofluorescence imaging

U2OS cells were plated on coverslips in 6-well plates and allowed to adhere overnight before being infected with wtAAV2 at the indicated MOIs for 24 hours. EdU labelling was carried out using 10 mM EdU stock solution diluted to a final concentration of 20 µM. The cells were pulsed with EdU for 2 hours. After the 2-hour incubation the samples were fixed using 4% PFA for 15 minutes at room temperature. Then the cells were washed with PBS and permeabilized with 0.5% Triton X-100 for 20 minutes at room temperature. After the incubation wash with PBS, 500 µL of the Click-it Reaction Cocktail (containing 1X Click-iT EdU reaction buffer, Copper sulfate, Alexa-Fluor-488 Azide and EdU buffer additive) was added to each sample and incubated for 30 minutes. Samples were washed with PBS. The samples were stained using 3% BSA in PBS (blocking) followed by incubation with the primary antibody for 30 minutes at room temperature. Samples were washed with PBS. Samples were stained with fluorophore conjugated secondary antibody in 3% BSA in PBS for 30 minutes in the dark at room temperature. Samples were washed in PBS mounted onto coverslips using DAPI-containing Fluromount. The samples were imaged on a Leica confocal microscope with 63X oil immersion objective.

### RNA extraction

Cells were plated and infected for 24 hours. The cells were then harvested and resuspended in 1 mL of PureZOL (Bio-Rad). 200 µL of Chloroform was then added to the microcentrifuge tubes and shaken vigorously for 15 seconds each. The samples

were then incubated for 5 minutes at room temperature on a rotator. The samples were centrifuged at 12,000xg for 15 minutes at $4^0$C. The aqueous layer was collected and transferred into a new microcentrifuge tube. The cells were precipitated with 2 µL of glycogen and 500 µL of isopropyl alcohol at minus 20 degrees C. Samples were centrifuged at 12,000xg for 10 minutes in $4^0$C. The RNA pellet was washed in 1 mL of 75% ethanol, centrifuged at maximum speed for 5 minutes at $4^0$C and air dried for 5 minutes. After the pellet was dried DNase treatment (Promega) was performed and incubated for 30 minutes at $37^0$C. The samples were then incubated at $65^0$C for 10 minutes to inactivate DNase. DNase stop solution was added to the sample. The RNA pellet was resuspended in PureZOL (Bio-Rad) and the purification was repeated. The pellet was resuspended in 20 µL of water. Reverse Transcription (RT) reaction was carried on 1 µg of RNA using the iScript cDNA synthesis kit (Bio-Rad). The generated cDNA was diluted in 80 µL of water and processed using qPCR.

## Western blots

Cells were plated in 6 well plates at 500,000 cells and infected with wtAAV2 virus for 24 hours. After the 24 hours cells were harvested and processed by lysis in RIPA buffer (containing protease inhibitors, sodium orthovanadate and sodium fluoride) on ice for 15 minutes. The cells were centrifuged at 13,000 rpm for 10-minutes at $4^0$C, supernatants collected and denatured with 6X Loading dye. Approximately 40 micrograms of lysate [measured using BCA assay (Bio-Rad)] were loaded on SDS-PAGE gels at 80-150V. Samples were transferred to a PVDF membrane using semi-dry transfer conditions at 25V for 30 minutes. The membrane was blocked with 5% milk in TBST for 30 minutes, incubated with appropriate primary antibody (1:1000, 1:2500, or 1:5000 dilution) and relevant secondary antibody (1:5000 dilution) for 1 hour each. Samples were incubated with ECL reagent (Bio-Rad) for 5 minutes and imaged on the LiCOR instrument.

## Supporting information

**S1 Fig. wtAAV2 infection induces replication stress in A549 and U2OS cells.** (A) Schematic of DNA fiber analysis using CldU pulsing of (B) A549 and (C) U2OS cells at 24 hpi when infected at an MOI of 5,000 vg/cell. Data is represented as median of 2 independent experiments with at least 100 datapoints per sample per replicate. Statistical analysis was performed using Mann Whitney Wilcoxon test with p value being depicted by ****, $p < 0.0001$.
(TIFF)

**S2 Fig. pAAV2 transfection induces replication stress and cellular DNA damage response signals.** (A) Schematic of pUC18/pAAV2 transfection of 293T cells followed by DNA fiber analysis at 24 hpi. Cells were transfected with 5,000 plasmid genome equivalents or 25,000 plasmid genome equivalents per cell. The resulting impact on host replication fork elongation was monitored by (A) DNA Fiber Analysis and (C,D) the impact on host DDR signals using western blots. DFA data is represented as median of 2 independent biological replicates. The levels of REP 68/78 protein in pAAV2 transfected cells were monitored by western blot.
(TIFF)

**S3 Fig. Verification of RNAi knockdowns.** The knockdowns of (A) SIK3, (B) XRCC1, (C) TFDP2 and (D) PRIM1 in 293T cells were verified by western blots at 24 hours.
(TIFF)

**S4 Fig. Impact of wtAAV2/rAAV2/scAAV2 on host genome stability.** The impact on wtAAV2 infection on host genomes were monitored by gamma H2AX western blots upon infection of (A) A549 and (B) HepG2 cells with the indicated viral genomes per cell for 24 hours with tubulin levels as loading control. The impact of rAAV2 and scAAV2 transduction on host genome stability of 293T cells was monitored by gamma H2AX western blots using the indicated MOIs of (C) rAAV2 and (D) scAAV2 vectors.
(TIFF)

**S5 Fig. Imaging of rAAV2/scAAV2 genomes relative to cellular DDR factors.** Immuno-FISH assays were performed to monitor to relative location of (A) rAAV2 and (B) scAAV2 genomes with that of the indicated phosphorylated DDR markers. DAPI staining was used to demarcate with nuclear borders with dashed lines and scale bars represent 10 microns. (TIFF)

## Acknowledgments

We thank all members of the Majumder Lab for insightful discussions and Rhiannon R. Abrahams (Majumder Lab, UW Madison) for expert technical guidance in generating AAV2 viruses. We thank Dr. David Pintel (University of Missouri) for scAAV2-GFP vector and Dr. Paul Lambert (UW Madison) for critical reading of the manuscript. We thank Dr. Robert Kalejta (UW-Madison) for NHDF cells, Dr. Marta Gaglia and Dr. Andrew Mehle (UW-Madison) for A549 cells and Dr. Dan Loeb (UW-Madison) for HepG2 cells. We thank Dr. Ci Ji Lim (UW-Madison) for PRIM1 antibody.

## Author contributions

**Conceptualization:** Monnette F. Summers, MegAnn K. Haubold, Clairine I. S. Larsen, Joshua J. Coon, Kavi P. M. Mehta, Kinjal Majumder.

**Data curation:** Monnette F. Summers, Kinjal Majumder.

**Formal analysis:** Monnette F. Summers, MegAnn K. Haubold, Marcel Morgenstern, Clairine I. S. Larsen, Ava E. Bartz, Gopishankar Thirumoorthy, Kinjal Majumder.

**Funding acquisition:** Kinjal Majumder.

**Investigation:** Monnette F. Summers, MegAnn K. Haubold, Marcel Morgenstern, Phoenix Shepherd, Clairine I. S. Larsen, Ava E. Bartz, Gopishankar Thirumoorthy, Kavi P. M. Mehta, Kinjal Majumder.

**Methodology:** Monnette F. Summers, MegAnn K. Haubold, Marcel Morgenstern, Phoenix Shepherd, Clairine I. S. Larsen, Ava E. Bartz, Kavi P. M. Mehta, Kinjal Majumder.

**Project administration:** Robert N. Kirchdoerfer, Joshua J. Coon, Kinjal Majumder.

**Resources:** Joshua J. Coon, Kinjal Majumder.

**Software:** Marcel Morgenstern, Joshua J. Coon.

**Supervision:** Robert N. Kirchdoerfer, Joshua J. Coon, Kavi P. M. Mehta, Kinjal Majumder.

**Validation:** Monnette F. Summers, Clairine I. S. Larsen, Kinjal Majumder.

**Visualization:** Monnette F. Summers, Gopishankar Thirumoorthy, Kavi P. M. Mehta, Kinjal Majumder.

**Writing – original draft:** Monnette F. Summers, Kavi P. M. Mehta, Kinjal Majumder.

**Writing – review & editing:** Clairine I. S. Larsen, Kavi P. M. Mehta, Kinjal Majumder.

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
