## [Decision Letter · Decision Letter 0]

27 May 2025

PPATHOGENS-D-25-00943

Adeno-Associated Virus 2 (AAV2) - induced RPA exhaustion generates cellular DNA damage and restricts viral gene expression

PLOS Pathogens

Dear Dr. Majumder,

Thank you for submitting your manuscript to PLOS Pathogens. After careful consideration, we feel that it has merit but does not fully meet PLOS Pathogens's publication criteria as it currently stands. Therefore, we invite you to submit a revised version of the manuscript that addresses the points raised during the review process.

Please submit your revised manuscript within 60 days Jul 26 2025 11:59PM. If you will need more time than this to complete your revisions, please reply to this message or contact the journal office at plospathogens@plos.org. Please include the following items when submitting your revised manuscript:

We look forward to receiving your revised manuscript.

Kind regards,

Cary A. Moody

Academic Editor

PLOS Pathogens

Blossom Damania

Section Editor

PLOS Pathogens

 Sumita Bhaduri-McIntosh

Editor-in-Chief

PLOS Pathogens

orcid.org/0000-0003-2946-9497

 Michael Malim

Editor-in-Chief

PLOS Pathogens

orcid.org/0000-0002-7699-2064

**Journal Requirements:**

At this stage, the following Authors/Authors require contributions: Monnette F Summers, MegAnn K Haubold, Marcel Morgenstern, Phoenix Shepherd, Clairine IS Larsen, Ava E Bartz, Gopishankar Thirumoorthy, Robert N Kirchdoerfer, Joshua J Coon, Kavi PM Mehta, and Kinjal Majumder. Please ensure that the full contributions of each author are acknowledged in the "Add/Edit/Remove Authors" section of our submission form.

- TM on page: 23.

5) We have noticed that you have uploaded Supporting Information files, but you have not included a list of legends. Please add a full list of legends for your Supporting Information files after the references list.

6) Please ensure that the funders and grant numbers match between the Financial Disclosure field and the Funding Information tab in your submission form. Note that the funders must be provided in the same order in both places as well.

**Reviewers' Comments:**

Reviewer's Responses to Questions

**Part I - Summary**

Reviewer #1: This study provides valuable insights into the mechanisms by which wtAAV2 monoinfection or rAAV2 transduction induce replication stress and activate the DNA Damage Response (DDR) in host cells (HEK293T and U2OS), with a focus on the critical role of RPA32 in these processes. The findings reveal a dose-dependent relationship between AAV2 infection and cellular replication stress, offering a deeper understanding of the interactions between viral genomes and host cellular pathways. Overall, the research provides meaningful contributions to our knowledge of AAV genome-induced DDR, which holds important implications for the use of rAAV in gene therapy.

Reviewer #2: In this manuscript, Summers et al characterize the ability of non-replicative vs replication competent AAV to induce the DNA damage response using a battery of single cell imaging and proteomics based analyses. The study highlights the ability of AAV genomes to deplete the ssDNA binding protein RPA in host cells, which leads to cellular stress and replication fork shortening, ultimately triggering a cascade of DNA damage related events. Chemical/RNAi/ectopic expression of RPA de-represses the AAV genomes/alleviates cellular stress. Overall, the conclusions are precise and focused, the study rigorously executed and the manuscript well-written. Some concerns/recommendations are outlined below.

Reviewer #3: Summers et al. present a well written and well-presented study in which they describe a mechanism of AAV2 inducing DDR signals and eventual cellular DNA damage in the host cell. They show that AAV2 associates with host RPA, which is involved in DNA replication and repair. They propose a model in which competitive binding by AAV2 genomes leads to RPA exhaustion on the host genome. While their model is convincing, some specific controls would make their conclusions better warranted and significantly improve the paper. Further, although individual sections are well presented, the connection between them is less-well described, making it difficult to understand why certain experiments were carried out.

**Part II – Major Issues: Key Experiments Required for Acceptance**

Reviewer #1: Major points:

1. Fig. 2 demonstrated that wtAAV2, rAAV2, and scAAV2 induce similar levels of replication stress. Fig. 5 showed that wtAAV2-mediated RPA32 exhaustion leads to cellular DNA damage. However, it is unclear whether rAAV2 can cause similar DNA damage, which could be explored by examining markers like rH2AX induction and colocalization of the AAV genome with DNA damage molecules (e.g., Fig. 5H). It is essential to test both single-stranded (rAAV) and double-stranded (scAAV) genomes, as they are used as viral vectors at high MOI. Since scAAV has a double-stranded genome, it may prevent RPA binding, which should be further examined.

2. Testing on additional cell types in Fig. 5 would strengthen the study's conclusions and expand its relevance to rAAV use in gene therapy applications.

Reviewer #2: All the DNA fiber experiments and CHIP experiments were performed in HEK293 cells, whereas the replication fork experiments were performed in U2OS cells. Is there are a reason behind this that the authors want to articulate? This distinction is important since HEK293 cells express Adenoviral E1A which is a ssDNA binding protein involved in host factor recruitment. When taken together with T antigen, there may be effects that require further deconvoluting if there is involvement of such factors. An additional experiment or two to demonstrate E1A/T antigen do not alter the conclusions would be useful.

In Figure 2, the difference between rAAV and AAV2 is clear. Additional controls to the short ITR oligo would be transfection of AAV genomes without ITRs (from plasmid) and/or transfected extracted AAV genomes (from full particles, but without capsid).

For Figure 4 and 5, were there additional experiments carried out with rAAV? While 4H shows some of this (protein read out), all AAV2 data is based on transcripts/epigenetic mods. The role of RPA2 here is also somewhat confusing here - as rAAV2 may not respond to RPA (Fig 4), but seems to have the same DNA damage response as AAV2 (Fig 2)?

Reviewer #3: - How specific is the damage induced by such high numbers of AAV2 genomes? Would transfecting the same amount of plasmid DNA also cause such reactions in the cell? A negative control of transfection 20,000 copies of plasmid DNA around the same size as the AAV2 genome would be a good comparison. In the same vein, is 20,000 copies of an AAV2 genome a physiologically relevant amount? This reviewer was somewhat disappointed that the lack of effect of rAAV2 was not further discussed since the introduction suggested that recombinant AAV2 vectors are important in the clinic, however, when no result was observed, further investigation was not carried out. It would be helpful to the reader to discuss the results in the context of how AAV is actually used in the clinic or infects cells, but these points are not discussed. Instead, the context of other viruses and the DDR is well described, perhaps even to the point where it seems less relevant to this study.

- It would be good to include a positive control for RPA subunits in fig 4a, meaning a region of the host that they should be bound.

**Part III – Minor Issues: Editorial and Data Presentation Modifications**

Reviewer #1: Minor points:

1. For better readability and consistency, AAV should be labeled as wtAAV2 throughout the manuscript.

2. Lines 168-170 contain a suggestion that is not well-supported by the data; this statement should be modified to align with the observed results.

3. In Fig. 3E, the manuscript should include data showing the knockdown efficiency of XRCC1, SIK, PRIM1, and TFDP2.

4. Lines 285-286 state, “plasmids expressing the RPA subunits (pRPA) into 293T cells prior to AAV2 infection as schematized in Fig. 4D.” The level of RPA expression should be confirmed and shown via Western blotting.

5. Not sure if the iPOND MS original data need to be deposited in a database.

Reviewer #2: The distinction between AAV2 vs rAAV2, AAV vs MVM, and viral vs host DNA damage is not always clearly articulated, when interpreting findings. Some careful delineation would be helpful for readers.

The model in Figure 6 could be expanded to show how this affects AAV/rAAV gene expression and perhaps how this compares to other parvoviruses (such as MVM that this group has made significant contributions towards understanding in the past).

Reviewer #3: Clarity issues:

- AAV2 monoinfection induces replication stress in the absence of any helper or other virus. scAAV2 and rAAV2 are sufficient to cause replication stress, but neither the core nor VP or ITR alone are. This suggests that it is the combination of these components working together that causes the replisome stress. Stating this would be helpful to the reader. For example, it is unclear why after showing these data, the authors then went to iPOND – the logic should be better stated.

- With respect to the iPOND data, it is also not well justified why the authors only followed up on RPA, which they already suspected was Why not follow up on anything else? Logic is a bit unclear since RPA2 was already suspected to be important, as was CSNK1.

- When RPA is bound to the AAV2 genome, the authors observe some repression because when RPA is not bound to the AAV2 genome, as per 4C, this results in greater gene expression, as per 4G (if these have the same copy numbers of genomes infected). However, this does not explain why the length of the replisomes is not different. Can the authors please explain this?

- The idea of RPA exhaustion is interesting but needs to be better explained. Does inhibition lead to exhaustion? Can inhibition be reversed by removal of the inhibitor?

- In fig 4I vs 4J – does the transfection of pRPA in 4J lead to a decrease in length with CIdU compared to IdU? It appears that is lower but no significance is noted.

- The logic of U2OS vs 293T cells is a bit unclear. 293T cells still express some adenovirus proteins albeit not the ones that would promote AAV2 infection, but why the authors switch from one cell type to the other is not well explained or justified.

- The interpretation that the MRN proteins co-localize with AAV2 genomes does not mean necessarily mean this is where the replisome stress is occurring – the resolution of the imaging doesn’t give you that proximity. It is likely but these caveats should be noted. Again the concern of cell type changes is unclear.

- The introduction and conclusion seem somewhat long winded without addressing some key points that should be discussed. For example, the points above with respect to cell types and caveats of interpretation.

- It is not clear why inhibition of RPA phosphorylation would prevent its binding to the AAV2 genome? And even when it is not bound, the replisome length is not affected in 4E and 4F comparing AAV2 to AVV2+inhibitor there is no significant difference. This could be better explained.

- The authors switch between mono-infection or monoinfection

- Line 206-207 – typo with the word ‘be’

- Line 310 – arising should be arise

- Line 404 – underpinnings is misspelt

PLOS authors have the option to publish the peer review history of their article (what does this mean? ). If published, this will include your full peer review and any attached files.

**Do you want your identity to be public for this peer review?** For information about this choice, including consent withdrawal, please see our Privacy Policy .

Reviewer #1: No

Reviewer #2: **Yes: ** Aravind Asokan

Reviewer #3: No

**Figure resubmission:**
---

## [Editor Report · Decision Letter 1]

29 Jul 2025

Dear Dr. Majumder,

We are pleased to inform you that your manuscript 'Adeno-Associated Virus 2 (AAV2) - induced RPA exhaustion generates cellular DNA damage and restricts viral gene expression' has been provisionally accepted for publication in PLOS Pathogens.

Best regards,

Cary A. Moody

Academic Editor

PLOS Pathogens

Blossom Damania

Section Editor

PLOS Pathogens

Sumita Bhaduri-McIntosh

Editor-in-Chief

PLOS Pathogens

orcid.org/0000-0003-2946-9497

Michael Malim

Editor-in-Chief

PLOS Pathogens

orcid.org/0000-0002-7699

---

## [Editor Report · Acceptance letter]

Dear Dr. Majumder,

We are delighted to inform you that your manuscript, " 

Adeno-Associated Virus 2 (AAV2) - induced RPA exhaustion generates cellular DNA damage and restricts viral gene expression," has been formally accepted for publication in PLOS Pathogens.

Best regards,

Sumita Bhaduri-McIntosh

Editor-in-Chief

PLOS Pathogens

orcid.org/0000-0003-2946-9497

Michael Malim

Editor-in-Chief

PLOS Pathogens

orcid.org/0000-0002-7699-2064